# Implantable batteryless device for on-demand and pulsatile insulin administration

Seung Ho Lee[1,*], Young Bin Lee[2,*], Byung Hwi Kim[3], Cheol Lee[4], Young Min Cho[5], Se-Na Kim[2], Chun Gwon Park[1], Yong-Chan Cho[2] & Young Bin Choy[1,2,3]

Many implantable systems have been designed for long-term, pulsatile delivery of insulin, but the lifetime of these devices is limited by the need for battery replacement and consequent replacement surgery. Here we propose a batteryless, fully implantable insulin pump that can be actuated by a magnetic field. The pump is prepared by simple-assembly of magnets and constituent units and comprises a drug reservoir and actuator equipped with a plunger and barrel, each assembled with a magnet. The plunger moves to noninvasively infuse insulin only when a magnetic field is applied on the exterior surface of the body. Here we show that the dose is easily controlled by varying the number of magnet applications. Also, pump implantation in diabetic rats results in profiles of insulin concentration and decreased blood glucose levels similar to those observed in rats treated with conventional subcutaneous insulin injections.

[1] Institute of Medical & Biological Engineering, Medical Research Center, Seoul National University, Seoul 03080, Republic of Korea. [2] Interdisciplinary Program in Bioengineering, College of Engineering, Seoul National University, Seoul 08826, Republic of Korea. [3] Department of Biomedical Engineering, Seoul National University College of Medicine, Seoul 03080, Republic of Korea. [4] Department of Pathology, Seoul National University College of Medicine, Seoul 03080, Republic of Korea. [5] Department of Internal Medicine, Seoul National University College of Medicine, Seoul 03080, Republic of Korea.
* These authors contributed equally to this work. Correspondence and requests for materials should be addressed to Y.B.C. (email: ybchoy@snu.ac.kr).

All types of diabetes mellitus are characterized by insufficient insulin secretion[1]. Type 1 diabetes is characterized by absolute insulin deficiency and thus requires insulin injections covering both basal and prandial requirements[1]. Type 2 diabetes is a consequence of both insulin resistance and insulin deficiency[1]. However, insulin therapy is necessary for people with type 2 diabetes who have severe hyperglycemia or hyperglycemia that is inadequately controlled by oral agents[2]. Multiple daily injections (MDIs) and continuous subcutaneous insulin infusion (CSII) can provide plasma insulin profiles similar to normal physiology[3]. However, MDIs are often painful and erroneous, and CSII requires a bulky, external device[4].

Therefore, a more patient-friendly system with controlled delivery of insulin has been actively sought. After one-time implantation (or injection), such a system must permit pulsatile insulin release to mimic physiological secretion following food intake[5]. More importantly, this release pattern must be controlled in a patient-driven, on-demand manner from outside the body without invasive multiple skin punctures[6]. Stimulus-responsive biomaterials previously used to deliver insulin[7–9] are subject to ruptures, leaks or deformations that lead to a lack of reproducibility over multiple release cycles. More sophisticated, implantable microdevices operated by the principles of peristaltic actuation[10,11] electrochemical dissolution[12,13], electrothermal ablation[14,15], electrolysis[16,17] or piezoelectric actuation[18] often require electrical power supplies (for example, batteries) and electronic circuit components[19] and are thus too large for implantation. Moreover, when the battery expires, device explantation is inevitable and requires additional major surgery, making this method unsuitable for long-time insulin delivery.

Therefore, in this work, we developed an implantable pump enabled with patient-driven, on-demand insulin release and, most importantly, without electric power sources, thus allowing semi-permanent use after implantation. We designed the pump to be actuated by an externally applied magnetic field (that is, a magnetically driven pump (MDP)). We prepared the pump by simple assembly of magnets and constituent units. The pump comprises a drug reservoir and an actuator to store and infuse insulin, respectively. The actuator is equipped with a plunger and barrel, each possessing a magnet. The plunger in the actuator moves to release an accurate amount of insulin stored in the reservoir only when a magnetic field is applied outside the body. To our knowledge, this study is the first to describe a simple-assembly, fully implantable pump for batteryless, on-demand insulin infusion via a magnetic field. Our findings show that after implantation of MDP in diabetic rats, the profiles of plasma insulin concentration and blood glucose levels were similar to those treated with conventional subcutaneous (s.c.) insulin injection.

## Results

**MDP design and working principles.** We prepared the MDP by assembling the two distinct drug reservoir and actuator units, as shown in Fig. 1a (Supplementary Fig. 1). The drug reservoir unit formed the exterior of the MDP. In the reservoir body cover, a refill port was formed with a septum, cover and reservoir magnet of donut-shape ($M_R$). Inside the drug reservoir, we prepared the actuator unit, which, like a typical syringe, was composed of two subunits, the plunger and barrel. The plunger subunit was composed of a plunger cover, plunger magnet of coin shape ($M_P$) and plunger body. The barrel subunit was composed of a barrel cover, barrel magnet of donut-shape ($M_B$), barrel body and check valve. The refill port septum was composed of poly-dimethylsiloxane (PDMS) elastomer. All magnets and the check valve were used as obtained from commercial sources. All other

units were fabricated from polyurethane copolymer using a PolyJet 3D printer to ensure high reproducibility of their precise dimensions. The units were then assembled and seamlessly attached with medical epoxy to produce the MDP. Detailed assembly procedures are provided in the Methods and Supplementary Fig. 1.

Figure 1b presents an optical image of the MDP and the external device prepared in this work. The total volume of the MDP was approximately 7.2 ml (30 mm × 16 mm × 15 mm, $l \times w \times h$), in which a 3 ml volume was used as the drug reservoir. We filled the MDP with 1.2 ml of an aqueous solution of insulin, which was injected through the septum port into the drug reservoir (Supplementary Fig. 1f). The whole surface of the MDP was coated with Parylene C due to the biocompatibility of Parylene C after implantation[20]. A one-way check valve was installed in series with the outlet to prevent unwanted leakage of insulin and inflow of bodily fluid. A pen-type external device was prepared following the basic design and dimensions of a commercially available insulin pen[21]. A magnet ($M_E$) was installed at the tip of the pen, and a cover was installed on the tip when not in use (Fig. 1b).

Figure 1c shows the actuation principle of the MDP in this work (Supplementary Movie 1). Magnets were placed in the heads of the barrel and plunger (that is, $M_B$ and $M_P$, respectively) to face each other with opposite polarities and therefore attach to each other strongly enough to prevent accidental movement of the plunger (also see 'Theoretical assessment of magnetic forces' in Supplementary Information). Under this non-actuation condition, pressure does not develop in the barrel, and the check valve remains closed, preventing any leakage of the insulin solution. A magnet ($M_E$) was installed in the external device facing the plunger magnet ($M_P$) with opposite polarity. The magnetic field between $M_E$ and $M_P$ is stronger than the one between the magnets in the barrel and plunger (that is, $M_B$ and $M_P$, respectively). Thus, when the external magnet device is applied above the MDP (Fig. 1c (①)), the plunger detaches from the barrel and moves upward (Fig. 1c (②)) to aspirate drug solution into the barrel (Fig. 1c (③)). Then, when the external magnet device is removed (Fig. 1c (④)), the plunger moves downward due to attraction from the magnet in the barrel, increasing the pressure in the barrel (Fig. 1c (⑤)) to open the check valve and release the insulin solution from the MDP (Fig. 1c (⑥)). The plunger was not tightly inserted in the barrel but with a gap of 150 μm to allow for smooth and reproducible actuations. The magnet in the drug reservoir ($M_R$) is for localization of the refill port, similar to medical devices such as tissue expanders[22]. Although not visible after implantation, therefore, the exact location of the refill port can be determined using a magnet of opposite polarity outside the body. The refill port allows refill injection using a hypodermic needle without damaging the body of the MDP. The $M_E$ faces the $M_R$ with the same polarity and is therefore repelled, preventing potential mislocation of the $M_E$ on the refill port.

***In vitro* performance test.** To investigate the performance of the MDP, an in vitro drug release test was performed, where a gap of 1 mm was prepared between the external device and MDP to simulate the presence of the skin after implantation[23,24]. In this work, the MDP could be actuated with gaps of up to 3 mm with the external magnet ($M_E$) of 3,000 G (Supplementary Table 1). As shown in Fig. 2a (Supplementary Fig. 2a), the release amount per actuation, 0.81 ± 0.04 U per actuation, was highly reproducible even with repeating actuations. This result implies that the MDP infuses 7.4 μl of liquid per actuation. Therefore, because the drug reservoir contains 1.2 ml of insulin solution, the MDP allows at least 160 deliveries of insulin after implantation. With this high

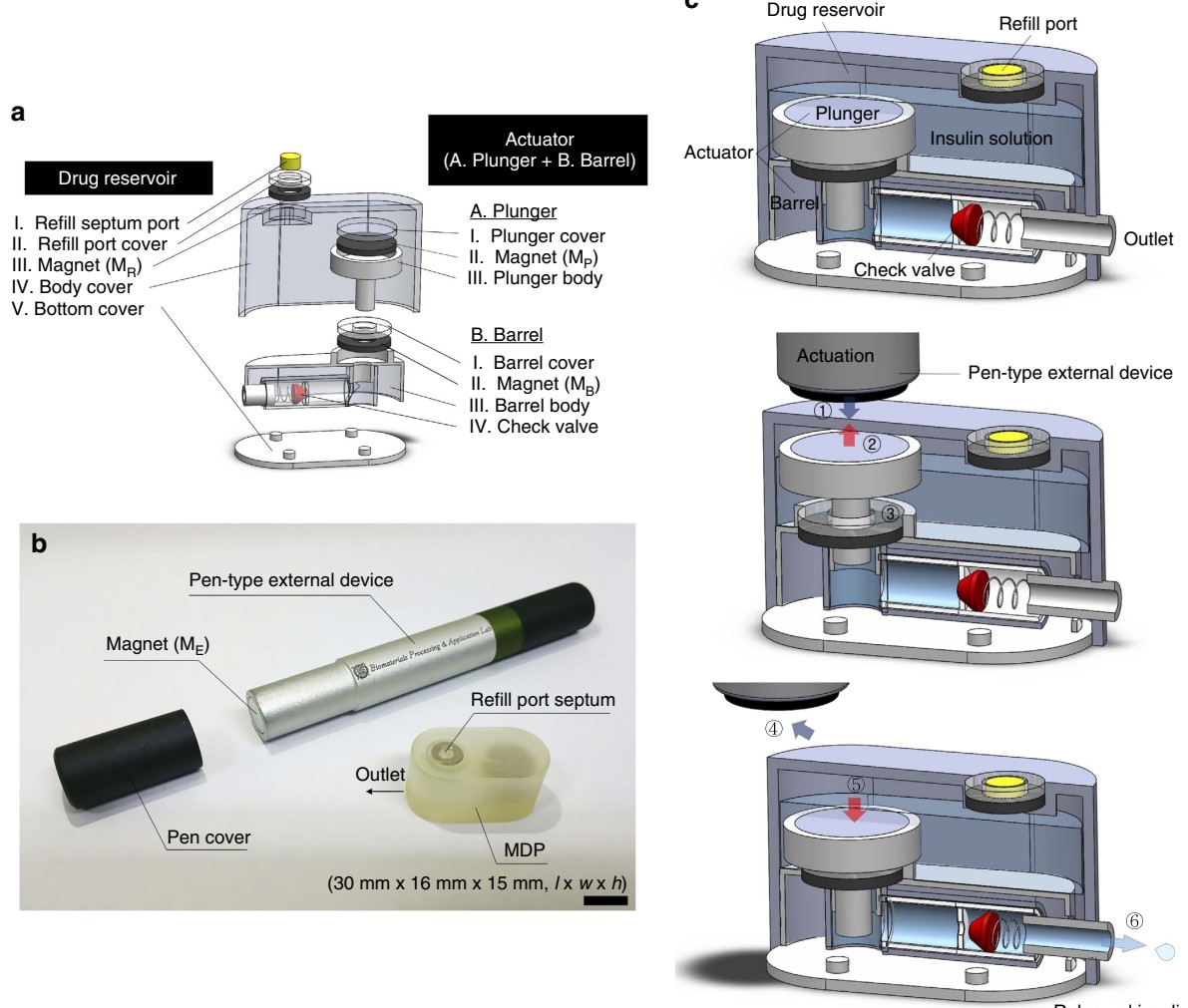

**Figure 1 | Descriptive images of the MDP.** (**a**) 3D schematic of the MDP (Solidworks, Dassault Systemes, USA). The MDP is composed of two distinct units: a drug reservoir and an actuator. The actuator is composed of a plunger and barrel. The drug reservoir is filled with insulin solution (109 U ml$^{-1}$) after assembly. To prepare the insulin solution, insulin in powder form (short acting; Sigma-Aldrich, MO, USA) was dissolved in sterile PBS at pH 7.4. (**b**) Optical image of the MDP and pen-type external device. Scale bar, 1 cm. (**c**) The actuation principle of the MDP (Solidworks, Dassault Systemes, USA) is as follows: ① the external magnet is applied; ② the plunger moves upward; ③ the drug solution in the reservoir is aspirated into the barrel; ④ the external magnet is removed; ⑤ the plunger moves downward; and ⑥ the insulin is released through the one-way check valve.

reproducibility, the dose amount can be varied by simply varying the number of actuations. To simulate conditions of long-term implantation, the MDP was fully submerged in release media for 28 days. As shown in Fig. 2b (Supplementary Fig. 2b), insulin was released only at the times of actuation, and no insulin was detected during the periods of non-actuation. Considering the LOQ (limit of quantification: 0.002 U ml$^{-1}$) with the measurement in this work, this result implied that even though there was a leak, it should be $<0.02$ U insulin during the longest non-actuation periods of 7 days ($<3 \times 10^{-3}$ U per day). The average amount of released insulin per actuation was similar to that described above, again indicating high reproducibility. Using the aqueous solution prepared in this work, the stability and biological activity of insulin appeared to be maintained for up to 30 days under simulated biological conditions (Supplementary Fig. 3).

***In vivo* evaluation**. To assess *in vivo* performance, the MDP was subcutaneously implanted in streptozotocin (STZ)-induced diabetic rats, and pharmacokinetics and pharmacodynamics studies were performed over a 30-day period. As shown in Fig. 3a, the insulin concentrations between the MDP_1A and s.c. injection groups were similar until day 11. The differences in concentration among the scheduled times were not significant for each of the animal groups. During this period, the insulin concentrations were 629.4 ± 12.3 µU ml$^{-1}$ and 683.3 ± 16.9 µU ml$^{-1}$ in the MDP_1A and s.c. injection groups, respectively, with a fairly narrow distribution of concentration values in both groups. The blood glucose levels and their decreases between the MDP_1A and s.c. injection groups were also similar until day 11 (Fig. 3b), with values of 238.1 ± 7.38 mg dl$^{-1}$ and − 251.1 ± 6.41 mg dl$^{-1}$, respectively. This result indicates that noninvasive actuation with a magnetic field from the outside skin allowed the implanted MDP to deliver insulin as effectively and reproducibly as the conventional s.c. injection. In the control group, both the insulin concentrations and decrements of blood glucose levels were very low (23.08 ± 9.08 µU ml$^{-1}$ and − 19.3 ± 7.57 mg dl$^{-1}$, respectively) during the testing period, as expected for STZ-induced diabetic

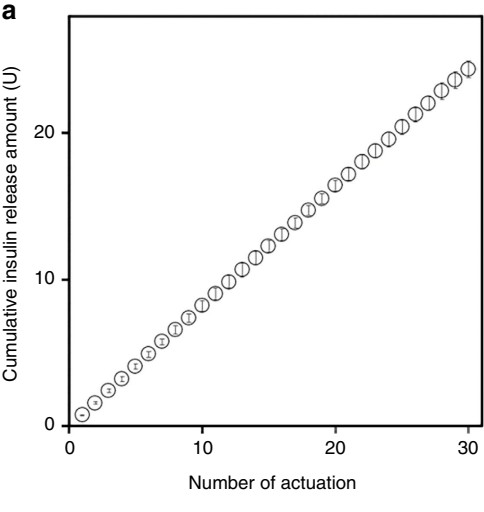

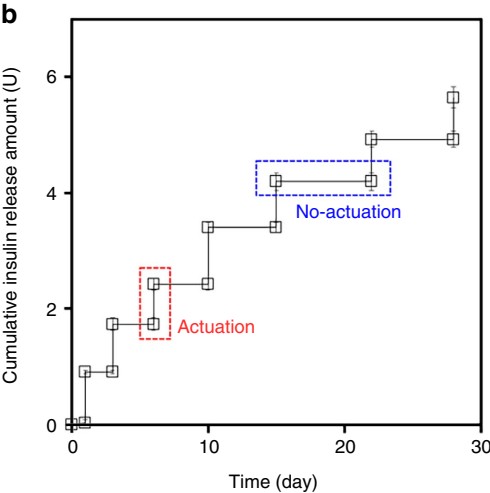

**Figure 2 | *In vitro* insulin release profiles of the MDP.** The MDP was fully immersed in pH 7.4 PBS and incubated in an incubator at 37 °C while being continuously shaken at 50 r.p.m. At each actuation, the external magnet device was applied at a constant distance from the top of the MDP by placing a 1 mm-thick glass slide between the external magnet device and MDP to simulate the presence of tissue after implantation, and removed almost instantaneously (<1 s) while the MDP was fully submerged in the release medium. An aliquot of the release medium was sampled at scheduled times and analysed by high-performance liquid chromatography. Details on this procedure are described in the Methods section. (**a**) Thirty consecutive actuations were applied at intervals of 10 min, and an aliquot was sampled after each of the actuations. With five distinct MDPs tested herein, the released amount of insulin was highly reproducible and was $0.81 \pm 0.04$ U per actuation. Error bars are s.d. (**b**) The MDP was actuated at predetermined times of 1, 3, 6, 10, 15, 22 and 28 days while fully immersed in PBS for 28 days. Aliquots were sampled immediately before and after each of the actuations. With four distinct MDPs tested herein, there was almost no release of insulin during the period of no actuation. The released amount of insulin was also highly reproducible and was $0.80 \pm 0.09$ U per actuation. Error bars are s.d.

rats[25]. After 11 days, the plasma insulin concentrations of the MDP_1A group decreased to as low as $430.9 \pm 22.8 \, \mu U \, ml^{-1}$ (Fig. 3a), which was associated with an attenuated blood glucose lowering effect (Fig. 3b). This decrease may be due to the formation of a fibrous capsule around the MDP after long-term implantation[15], which appeared to hamper the diffusion of insulin from the s.c. space into the blood stream.

To compensate for this reduction and attain therapeutic levels similar to the s.c. injection group, we proposed to deliver more insulin with the MDP after 11 days. Another animal group, MDP_1A/2A, was prepared to test this method. In this group, the MDP was actuated once until day 11; the insulin concentration and decreased glucose levels ($636.5 \pm 29.2 \, \mu U \, ml^{-1}$ and $-240.7 \pm 26.4 \, mg \, dl^{-1}$, respectively) were not very different from the MDP_1A and s.c. injection groups (Fig. 3). After 11 days, with two consecutive actuations at each administration, the MDP_1A/2A group exhibited an insulin concentration and decreased glucose level of $741.8 \pm 4.13 \, \mu U \, ml^{-1}$ and $-300.3 \pm 10.8 \, mg \, dl^{-1}$, respectively, both higher than in MDP_1A and, importantly, similar to the values in the s.c. injection group. During the non-actuation periods in the MDP_1A and MDP_1A/2A groups, plasma insulin concentrations were low (Supplementary Fig. 4) and similar to the control group, indicating that there was no leakage of insulin from the implanted MDP. To examine a long-term efficacy, we continued the experiment with the MDP_1A/2A group until 60 days, where the insulin in the MDP was fully replenished at 31 days. As shown in Fig. 3c,d, the insulin concentration and decreased glucose levels from 31 days ($743.9 \pm 10.9 \, \mu U \, ml^{-1}$ and $-323.8 \pm 19.9 \, mg \, dl^{-1}$, respectively) were similar to those observed during the earlier period at 16–30 days (Fig. 3a,b). This result implied a long-term applicability of the MDP herein that could infuse the insulin in a reproducible manner even after a replenishing procedure. After a replenishing procedure, we did not observe any sign of hypoglycemia with all tested animals. Due to the valve located at the outlet, a slight change in reservoir pressure did not appear to cause considerable insulin burst release.

For a more detailed analysis, we also assessed the dynamic profiles of insulin concentration and decreased glucose level at shorter time scales (that is, 720 min) after administration in the s.c. injection, MDP_1A and MDP_1A/2A groups. These extensive measurements were performed at 1, 16 and 30 days after MDP implantation. As shown in Fig. 4, on 1 day, the profiles of insulin concentration and decreased glucose levels were similar among all three groups. However, on days 16 and 30, the overall decrease in the insulin concentration and decreased glucose level were greater in the MDP_1A group than in the s.c. injection group. The areas under the curve for insulin concentration and decreased glucose level ($AUC_{PK, \, insulin}$ and $AUC_{PD, \, glucose}$, respectively) in the s.c. injection group were $84,392 \, \mu U \, ml^{-1}$ min and $60,035 \, mg \, dl^{-1}$ min, respectively, which decreased to $50,236 \, \mu U \, ml^{-1}$ min and $41,325 \, mg \, dl^{-1}$ min on day 16 and $47,162 \, \mu U \, ml^{-1}$ min and $36,510 \, mg \, dl^{-1}$ min on day 30, respectively. In the MDP_1A/2A group, the maximum insulin concentration and decrease in glucose level were comparable to those in the s.c. injection group on days 16 and 30. However, both $AUC_{PK, \, insulin}$ and $AUC_{PD, \, glucose}$ were larger in the MDP_1A/2A group ($129,204 \, \mu U \, ml^{-1}$ min and $93,364 \, mg \, dl^{-1}$ min at day 16 and $129,267 \, \mu U \, ml^{-1}$ min and $90,360 \, mg \, dl^{-1}$ min at day 30, respectively) than in the s.c. injection group. In the MDP_1A/2A group, a larger dose of insulin was administered into the s.c. space and appeared to slowly diffuse into the blood stream across the fibrotic capsule around the MDP, resulting in greater systemic insulin exposure. This process appeared to be completed within hours after MDP actuations (Supplementary Table 2). Regardless of the period after implantation, the times of the maximum insulin concentration and decrease in glucose level were constant (that is, $T_{max, \, insulin} = 60$ min and $T_{max, \, glucose} = 120$ min, respectively) in all groups.

**Histopathology.** To assess *in vivo* biocompatibility, biopsied tissue samples around the MDP were examined by hematoxylin

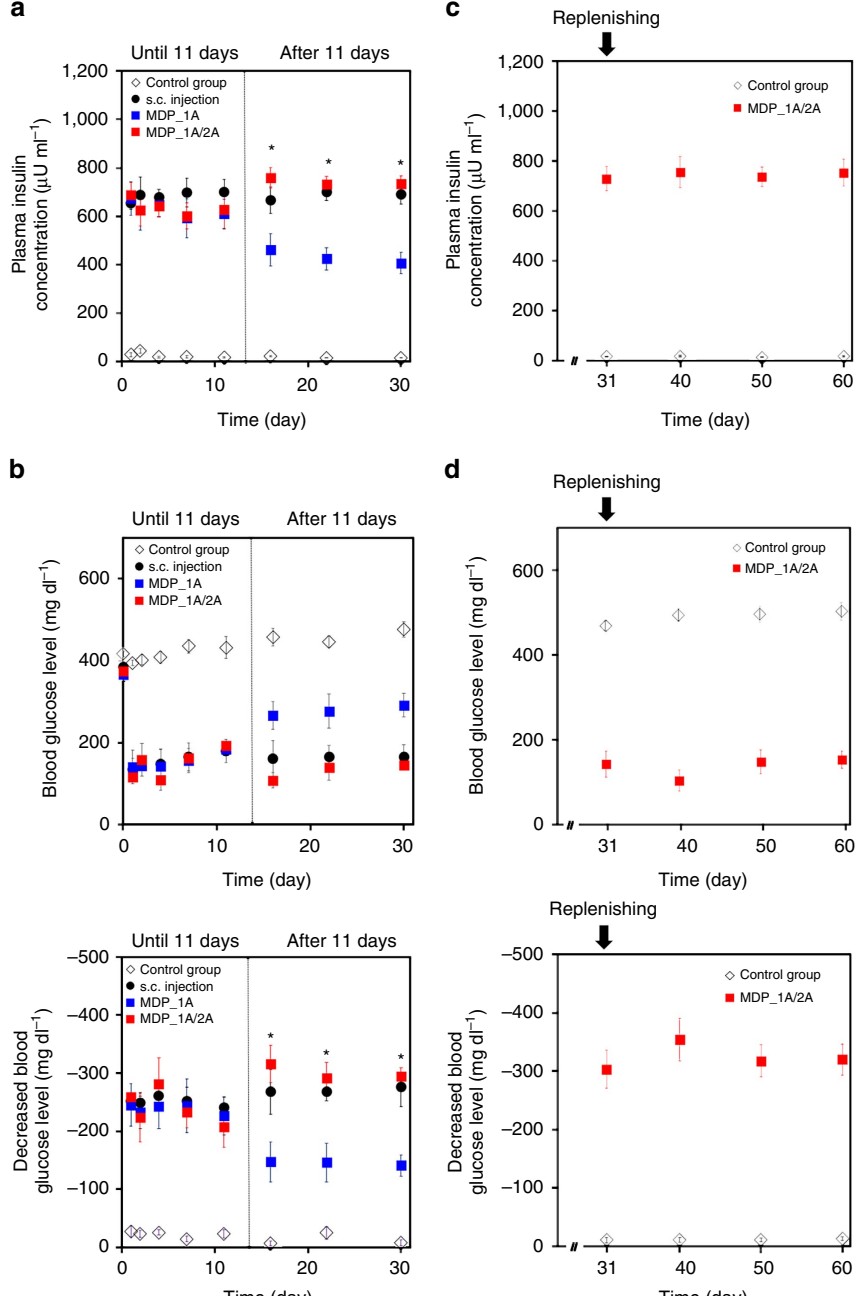

**Figure 3 | *In vivo* profiles of insulin delivery.** The profiles of (**a,c**) plasma insulin concentration and (**b,d**) blood glucose level were obtained from the four different animal groups: (i) control group-diabetic rats with no treatment ($n = 4$), (ii) s.c. injection group-diabetic rats subcutaneously injected with an insulin solution (0.8 U insulin) with a Hamilton microlitre syringe at each of the scheduled times ($n = 4$), (iii) MDP_1A group-diabetic rats implanted with the MDP in the subcutaneous space and treated with a single actuation (0.8 U insulin) at each of the scheduled times ($n = 4$); and (iv) MDP_1A/2A group-diabetic rats implanted with the MDP and treated with a single actuation (0.8 U insulin) at each of the scheduled times until 11 days and two consecutive actuations (1.6 U insulin) at each of the scheduled times after 11 days ($n = 4$). For each actuation, the external device with $M_E$ was applied and removed to the skin immediately above the implanted MDP. After insulin was delivered via actuation or injection, blood was withdrawn at $T_{max, insulin} = 60$ min to measure the maximum insulin concentration and also at $T_{max, glucose} = 120$ min to measure the minimum glucose level and its maximum decrease (Supplementary Fig. 11). Error bars are s.d. (**a,b**) The plasma was sampled at scheduled times of 1, 2, 4, 7, 11, 16, 22 and 30 days with the four different animal groups. (**c,d**) After 30 days, we continued the experiment with the two different animals groups: (i) control and (iv) MDP_1A/2A groups. At 31 days, for the MDP_1A/2A group, we fully withdrew the insulin solution in the drug reservoir and refilled it with 1.2 ml of a fresh one while the MDP was still implanted (Supplementary Fig. 6). The plasma was sampled at scheduled times of 31, 40, 50 and 60 days. *, the MDP_1A group was statistically significantly different from the MDP_1A/2A group ($P < 0.05$).

and eosin (H&E) and CD68 staining. As shown in Fig. 5a,b, the overall inflammatory and foreign body reactions were minimal for all tested tissue locations around the MDP. Also, elevation of inflammatory markers in plasma, such as interleukin (IL)-1b, IL-6

and tumour necorosis factor-α, was not observed at 30 and 60 days after MDP implantation (Supplementary Table 3).

These results support the biocompatibility of the MDP, which is attributable to the biocompatible Parylene C coating

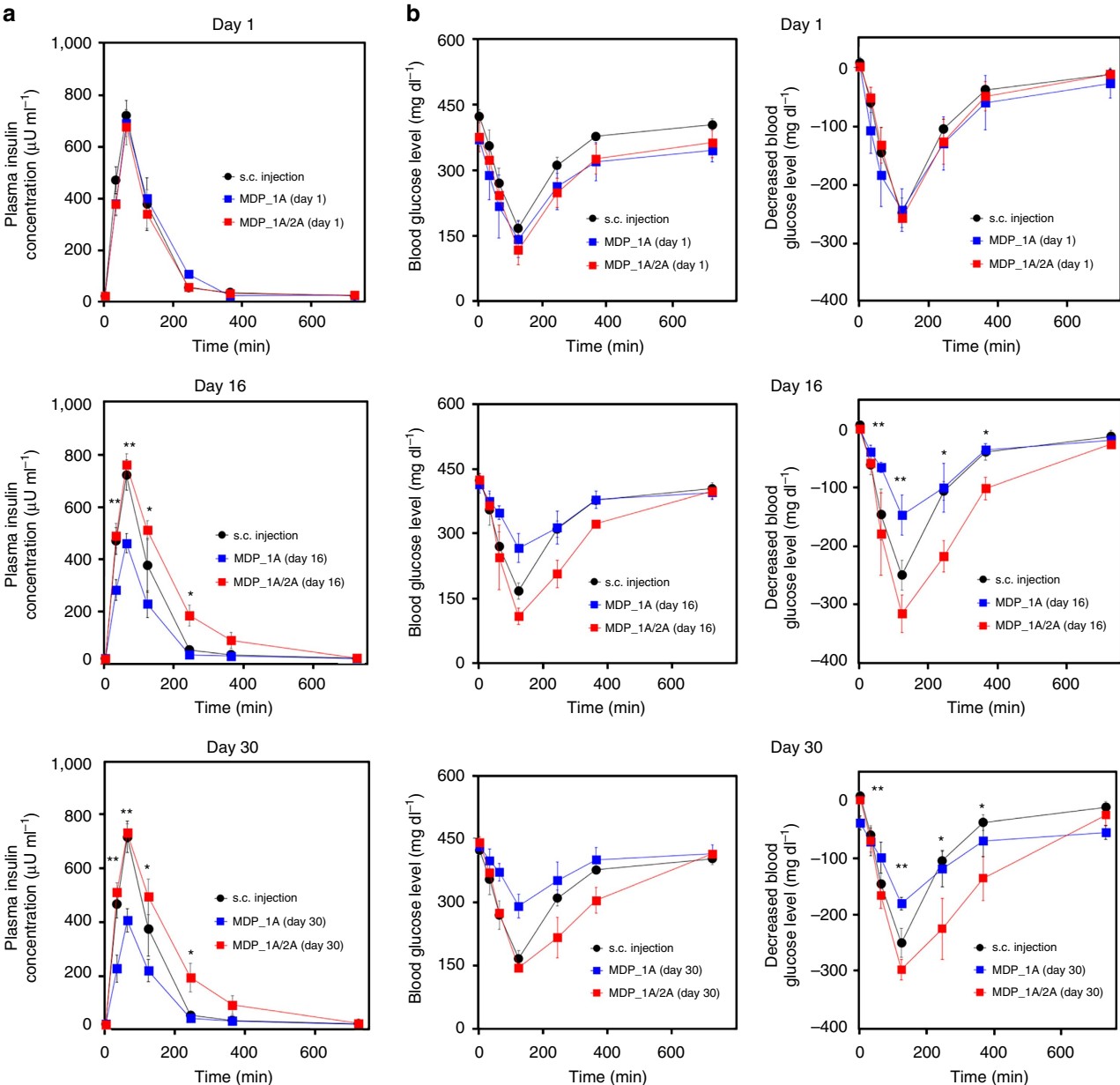

**Figure 4 | In vivo short-time profiles of insulin delivery.** The profiles of (**a**) plasma insulin concentration and (**b**) blood glucose level were obtained at shorter time scales at -1 to 720 min on days 0, 16 and 30 from the four different animal groups: (i) control group ($n=4$), (ii) s.c. injection group ($n=4$), (iii) MDP_1A group ($n=4$) and (iv) MDP_1A/2A group ($n=4$). Error bars are s.d. **, the MDP_1A group was significantly different from both the s.c. injection and MDP_1A/2A groups ($P<0.05$). *, the MDP_1A group was statistically significantly different from the MDP_1A/2A group ($P<0.05$).

used in this work[20]. The fibrotic capsule close to the outlet of the MDP, which could influence the pharmacokinetics and pharmacodynamics profiles of infused insulin, was also examined. As shown in Fig. 5c, fibrotic capsule formation was evident, and in the capsule tissues, minimal proliferation of bland-looking fibroblasts with rich collagenous stroma and some lymphoplasmacytic, histiocytic and eosinophilic infiltration were observed. The capsule thickness was $242 \pm 56.9\,\mu m$ after 16 days, similar to the thicknesses at 30 and 60 days ($243 \pm 56.9$ and $246 \pm 41.7\,\mu m$, respectively). No sign of clogging was observed with the valve in the MDP at 60 days (Supplementary Fig. 5). This constant thickness may explain the reproducible values of insulin concentration and decreased glucose level measured from 16 days in the MDP_1A and MDP_1A/2A groups. The capsule thickness did not appear to vary at the different locations of the MDP surface.

## Discussion

In this work, we designed a fully implantable MDP for semi-permanent use. Compared to previous implantable systems for insulin delivery[11,14,26], our MDP is advantageous due to its batteryless operation by infusing insulin based on actuation via a magnetic field. The magnetic field strength used in this study ($\sim 0.1\,T$) can propagate through tissue without damage[27], and thus, the MDP can be actuated noninvasively. This approach also allows the MDP to be patient-driven to provide on-demand release of insulin. This is an important aspect of clinical modulation for maintaining blood glucose levels after each meal[28]. Along with the batteryless design, we also prepared a refill port in the drug reservoir to enable periodic refilling of insulin. The MDP exhibited similar infused insulin doses per actuation before and after the refilling procedure, and thus the *in vivo* pharmacokinetics and pharmacodynamics profiles did almost not

change after a refilling procedure (Fig. 3; Supplementary Fig. 6). Similar to tissue expanders in clinical use[22], the locations of the refill port can be easily recognized due to the presence of a magnet, $M_R$, even with s.c. implantation of the MDP. In this work, to examine the feasibility, we prepared the MDP as a prototype, small enough to be implantable in STZ-induced

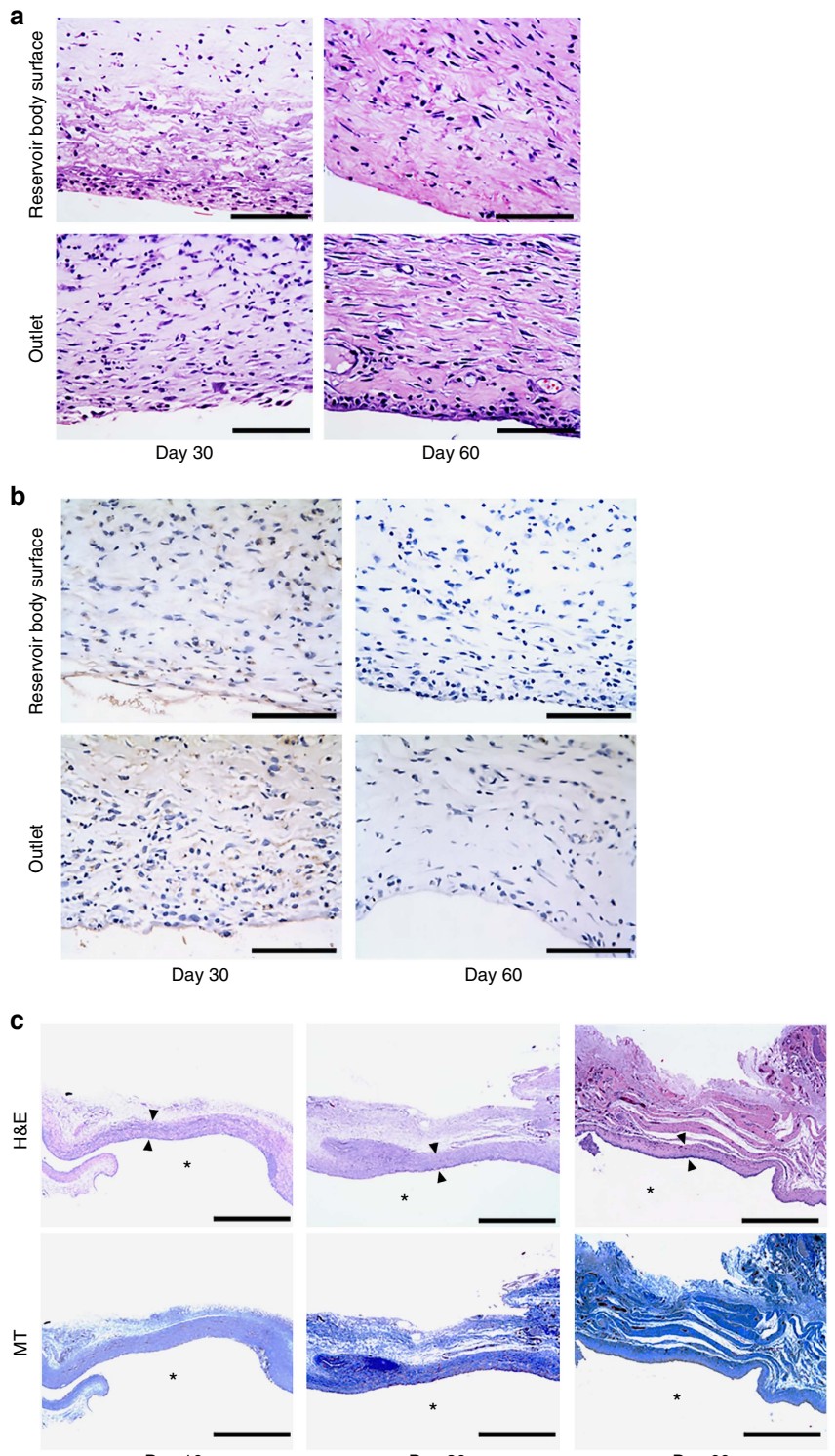

**Figure 5 | Representative histological images of the tissues around the MDP.** Two distinct locations in the tissue were observed: the tissues near the reservoir body surface and near the outlet of the MDP. The asterisk (*) indicates the location of the implanted MDP. $n = 4$; 16 days, $n = 5$; 30 days and $n = 4$; 60 days. To evaluate biocompatibility, we assessed the degree of inflammatory response in (**a**) hematoxylin and eosin (H&E)-stained and (**b**) CD68-stained tissues near both the reservoir body surface and the outlet of the MDP at 30 and 60 days after implantation. Scale bar, 100 μm. (**c**) Formation of collagen was assessed in both H&E- and Masson's trichrome (MT)-stained tissues near the outlet of the MDP at 16, 30 and 60 days after implantation. To measure the capsule thickness, the thinnest region of the capsule was selected in each image of the H&E-stained samples, as indicated by the arrows. Scale bar, 1 mm.

diabetic rats. Therefore, envisioning implantation in human, the MDP could be prepared to possess a larger volume, hence allowing for a larger drug reservoir. This could be designed more efficiently as the MDP herein does not need a space for battery.

Previous trials have employed magnetic fields to trigger drug release from implantable devices. For example, a drug reservoir in an implantable device was sealed with a porous membrane in which the pores could be opened and closed by an externally applied magnetic field[29]. In other examples, a membrane with an aperture or porous capsule was deformed to squeeze the drug depot and initiate its release by a magnetic field[30–32]. However, due to the presence of pores and apertures in the sealing membrane, drug leakage during non-actuation periods was not avoidable. This inherent drawback may hamper the application of these devices for insulin delivery because uncontrolled exposure of insulin can cause undesirable events, such as hypoglycemia[33]. We therefore designed the MDP to be leak-free when the external device is not in use. Two magnets embedded in the MDP become attached to immobilize the plunger and barrel subunits, preventing unwanted infusion. We also employed a check valve at the outlet to further prevent leakage possibly caused by diffusion (Supplementary Fig. 4).

Another advantage of our MDP is the relatively fast infusion of insulin, where the delivered doses of insulin per actuation were quite similar regardless of the period for the external device application (Supplementary Fig. 7). Upon removal of the external device, the magnet in the plunger immediately attracts the magnet in the barrel, causing the plunger to push insulin toward the outlet nearly instantaneously. Although not the same as the biological system, such as islets[34], this rapid infusion more closely mimics insulin delivery via the established clinical modalities, such as with an insulin pen or needle–syringe injections[35]. The pharmacokinetics and pharmacodynamics profiles of insulin release in the MDP group were similar to those in the s.c. injection group, even at shorter time scales (Figs 3 and 4). Therefore, the MDP herein could be applicable for a variety of insulin formulations and regimens to achieve a profile of blood glucose level needed for specific diabetic treatment (Supplementary Figs 8 and 9). Dose adjustment by varying the number of consecutive external device applications can also be easily performed because of the fast mechanical response between the plunger and barrel. At each actuation, the MDP infuses a specific volume of liquid, and therefore the dose of each actuation can also be varied by changing the concentration of insulin solution stored in the drug reservoir. Thus, we envision that a dose of prandial insulin can be customized within the typically prescribed range of 0.5–1.0 U, depending on the patient's needs for precise insulin dosing[36]. This dose adjustment would be more effective and convenient when combined with a closed-loop glucose sensor[37,38].

Even with the presence of fibrotic capsules, the MDP produced plasma insulin concentrations and decreased glucose levels similar to those in the early period after implantation, simply by increasing the number of actuations (Fig. 3). Although the overall inflammatory and foreign body reactions were acceptable (Fig. 5), fibrotic capsule formation around the nondegradable implant like the MDP was natural and inevitable[39]. Fibrotic capsule formation occurs gradually over weeks[40], and thus to ensure reproducible systemic drug exposure, we envision delaying the operation of the MDP herein until formation of the fibrous capsule is complete[15]. Another clinical issue is local lipohypertrophy at the site of s.c. insulin injection[41]. In this scenario, a catheter-connected MDP can be advantageous. For example, insulin can be infused via a catheter into the intraperitoneal space while implanting the MDP subcutaneously

for improved access to the external magnetic device[42] (Supplementary Movie 2; Supplementary Table 4).

In conclusion, we have proposed a simple-assembly, implantable infusion pump enabled with on-demand, pulsatile release of insulin. The pump can be actuated to infuse insulin via an externally applied magnetic field, thus enabling noninvasive insulin delivery after one-time implantation. Most importantly, the pump can be operated without a battery and can be designed to be refilled, enabling potentially semi-permanent use. The desired dose of insulin can be accurately and reproducibly controlled by varying the number of actuations, resulting in pharmacokinetics and pharmacodynamics profiles similar to those of conventional s.c. insulin injections. This control can be achieved simply by varying the number of applications of a pen-type magnet on the outside skin above the implanted insulin-infusion pump. Therefore, we conclude that the system proposed in this work is promising for noninvasive, on-demand pulsatile insulin administration for diabetic treatment.

## Methods

**MDP fabrication.** The three-dimensional structure of the MDP was drawn using 3D modelling software (SolidWorks, Dassault Systemes, USA). The composing units and their subunits were separately fabricated from polyurethane copolymer (Fullcure 720 photopolymer) using a rapid prototyping PolyJet technique on a 3D printer (Eden 350 V, Objet Geometries, Israel). The resulting units and subunits were then assembled with the magnets (∼0.1 T) and seamlessly attached with medical epoxy (Epo-Tek 301, Epoxy Technology, USA). A more detailed description of the MDP fabrication procedure is provided in Supplementary Fig. 1.

**Measurement of insulin concentration.** The insulin concentration was measured using an Agilent 1,260 series HPLC system (USA) equipped with a UV detector set at 210 nm and a reversed-phase poroshell 300SB-C18 column (5 μm, 75 mm × 2.1 mm i.d., Thermo Scientific, USA). The mobile phase was a mixture of acetonitrile with 0.08% trifluoroacetic acid (ACN/TFA) and 0.1% trifluoroacetic acid aqueous solution (TFA). The mobile phase was fed at a flow rate of 0.8 ml min$^{-1}$ in gradient mode, in which the volume ratio of ACN/TFA:TFA was 25:75 for the first 1.5 min and 40:60 for the remaining 5 min.

**In vivo experiments.** The protocol for in vivo experiments was approved by the Institutional Animal Care and Use Committee (IACUC No. 14–0165) at Seoul National University Hospital Biomedical Research Institute. To induce a diabetic animal model, male Sprague-Dawley (SD) rats weighing 350–400 g were fasted for 8 h with free access to water and then intraperitoneally injected with 60 mg kg$^{-1}$ streptozotocin (STZ; Sigma-Aldrich) dissolved in 0.09 M citrate buffer solution (pH 4.0) to destroy the insulin-producing beta cells of the pancreas[43]. Five days after STZ treatment, the rats were fasted for 8 h, and then approximately 5 μl of blood was sampled from the tail vein and analysed with a glucose metre (Accu-Chek Performa, Roche Diagnostics, Germany). Only rats with blood glucose levels >300 mg dl$^{-1}$ were employed as diabetic rats. The STZ-induced diabetic rats were divided into four groups: (i) control group; (ii) s.c. injection group; (iii) MDP_1A group; and (iv) MDP_1A/2A group. No randomization was used. Details on the surgical procedure for MDP implantation are described in Supplementary Fig. 10. During the experiments, the animals were freely accessible to food and water. To measure the plasma insulin concentration at each scheduled time, 0.5 ml of blood was withdrawn from the tail vein 60 min after insulin administration while the animal was sedated by inhalation of isoflurane. For the MDP_1A and MDP_1A/2A groups, blood was also withdrawn at −1 min (that is, at 1 min before MDP actuation), and the insulin concentration was measured to monitor any possible leak of insulin from the MDP (Supplementary Fig. 11). At 1, 16 and 30 days, blood was also withdrawn at shorter time scales at −1, 30, 60, 120, 240, 360 and 720 min after insulin administration (Fig. 4a). The blood plasma was separated by centrifugation and stored in the freezer at −20 °C before analysis. The insulin concentration in plasma was measured using an Insulin ELISA kit (Mercodia Insulin ELISA, Mercodia AB, Sweden). To measure the blood glucose levels at each scheduled time, ∼5 μl of blood was sampled at 120 min after insulin administration and analysed using a glucose metre. The decreased glucose level was calculated by subtracting the glucose level at 120 min from that measured at −1 min (that is, at 1 min before insulin administration). At 1, 16 and 30 days, the blood glucose level was also obtained at shorter time scales at −1, 30, 60, 120, 240, 360 and 720 min after insulin administration (Fig. 4b).

**Histopathology.** The rats in the MDP_1A/2A group were euthanized by carbon dioxide inhalation at 16 and 30 days after implantation, and the dorsal region tissues around the implanted MDP were harvested for histopathological analysis.

The tissues were fixed in 10% buffered formalin in a conical tube and embedded in paraffin wax for sectioning and staining. The paraffin blocks were then cut into 4-μm-thick slices and prepared on glass slides. The slides were then stained with hematoxylin and eosin (H&E), CD68 and Masson's trichrome (MT). Finally, the stained slides were assessed by a board-certified professional pathologist using an optical microscope (BX53, Olympus, Japan). No blinding was done. In this work, we observed tissue near the reservoir body surface and tissue near the outlet of the MDP. At each biopsy time and for each tissue location, at least three images were obtained from each of the four rats; a total of at least 12 images were analysed.

**Statistical analysis**. All data for plasma insulin concentration and decreased glucose level in blood at each scheduled time were compared among the s.c. injection, MDP_1A and MDP_1A/2A groups for statistical significance via non-parametric methods (Kruskal–Wallis ANOVA). $P < 0.05$ was considered statistically significant. The statistical software application used was SPSS (SPSS version 22, IBM, USA).

**Data availability**. Data supporting the findings of this study are available within this article and its Supplementary Information Files and from the corresponding author upon reasonable request.

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

## Acknowledgements

This work was funded by a grant from the Korea Healthcare Technology R&D Project, Ministry for Health, Welfare & Family Affairs, Republic of Korea (HI14C2194).

## Author contributions

Y.B.C. conceived and supervised the whole project. S.H.L. and Y.B.L. prepared and characterized the insulin infusion system with C.G.P. and Y.-C.C. S.H.L., Y.B.L., B.H.K. and S.-N.K. conducted in vivo experiments. C.L. performed histological analyses. Y.B.C., Y.M.C., S.H.L. and Y.B.L. wrote the manuscript.
