## [Peer Review File · Nature Communications]

Reviewers' Comments:

Reviewer #1 (Remarks to the Author)

The paper by Lee et al presents an interesting concept to use magnets in an implantable insulin pump to avoid the requirement for batteries. Any device that can improve the quality of life of diabetics, e.g. avoid multiple daily injections, decrease the incidence of hyper-/hypo-glycaemia, is welcome. The manuscript is clearly written. I have a few questions for the authors.

Major questions/comments:

1. Accurate activation and deactivation of the delivery mechanism is key. What is the maximum distance between the external and internal magnets before the mechanism stops working optimally? Are there any mechanisms to mitigate against accidental activation by a magnet in the vicinity?
2. Which insulin preparation was used in the test? It would be useful for application purpose to test different insulin preparations to check efficacy using the device as diabetics use different preparations.
3. What happens if there is insulin solution leftover in the reservoir that is degraded and therefore no longer efficacious? Can this be easily removed following implantation? The authors mention using 1.2 ml of insulin in the reservoir when the reservoir can hold 3 ml, i.e. theoretically ~400 actuations, what is the rationale for this?
4. How is the actuation volume controlled? By the amount of time the magnet is applied to the device? How envisage this applies to human use- calculations may be a bit too much for a patient?
5. Experimental details of the in vitro test were not given so it is difficult to visualize the experiment- e.g. how is distance of 4.5 mm from top of MDP to external magnet achieved? The authors state that there was almost no release of insulin in the period of no actuation; it would be useful to show the numerical data in the text. The error bars in Figure 2a, b are hard to see; how many devices were tested, how many times was this test repeated? Summary data showing average release with and without actuation would be useful, together with the variation as a separate figure.
6. In vivo tests were only performed for 30 days; in fig 3a MDP_1A (and possibly MDP_1A/2A) appear to be drifting downward (the drift also evident in 3b), and it would make sense to continue the experiment till this stabilises. Was the insulin in the reservoir continually replenished during this period and if so it would be useful to show glucose concentrations and plasma insulin content after replenishment over this time course? Please state the predetermined time of day of actuation; how was this time decided upon- feeding behavior? It is difficult to rationalize the protocol described in the in vivo experiment with the human situation where insulin is injected at meal times to normalize blood glucose. What happens if you need to actuate at different times of the day e.g. to mimic meal times?
7. The authors mention that there was potentially a gradual release from the fibrotic capsule which may affect plasma insulin content, leading to a modification of the protocol. The authors should assess how much insulin was present in the fibrotic capsule at day 11 and following the 1 and 2 actuation protocols so as to determine how much insulin can be stored in the fibrotic cap (essentially a second reservoir). Could this be the reason why the insulin content at day 16 and 30 is higher than SC and takes longer to return to baseline (Fig. 4a), which is mirrored in the blood glucose content in 4b? This is important due to the dangers of hypoglycaemia. It would be interesting to check fibrotic capsule size for longer than 30 days post implant to determine if the fibrotic capsule may get any thicker/larger and worsen the second reservoir effect. The authors should also perform more detailed analysis of the inflammatory response: staining for markers of inflammation in the sections and measurement of inflammatory markers in plasma should be performed.
8. The device releases insulin in pulses but it is not the same as the pulsatile release exhibited by islets and the manuscript needs to be carefully worded so the two phenomenon are not

confounded.

Minor comments:

9. It would help the readers if the size of the device is indicated in the figures (e.g. Fig 1).

Reviewer #2 (Remarks to the Author)

This paper presents a magnetically operated pump that is able to deliver insulin. Animal model was presented to show evidence of the working principle. Overall, this work, while provides value in engineering design and perhaps the field of bioengineering, is unlikely to inspire thinking for general audience of Nature Communications.

Below, please find suggestions to improve the manuscript.

Data & methodology

1. Line 128-129: "... As shown in Figure 2b, insulin was released only at the times of actuation, and no leakage was observed during the periods of non-actuation." The authors implies that the valve has zero leakage. Perhaps the leakage amount is undetectable?

2. Line 137, what is "S.C. injection group"? subcutaneous?

3. Will catheter length affect the amount of insulin delivering under the same magnetic field? Perhaps to due friction loss, more pressure from the plunger is required to push the same amount of fluid out as the catheter length increases. How would the present design to accommodate for that?

4. Line 28:"...implantable insulin pump that can be actuated by a static magnetic field.". The statement is not true, the magnetic field has to vary, otherwise, one would see no motion from the plunger. So a user uses a magnet to move the plunger, the speed of the user moves the magnetic towards or away from the plunger will affect the speed of the plunger, and the speed of the fluid, will this speed differences affect insulin delivery?

5. As the implantation period is longer than 11 days, insulin delivery goes down, the authors argue that the formation of a fibrous capsule could be to blame. It would be easy to just look at the catheter and determine for sure whether or not fibrous tissues is indeed the cause. Can the authors rule out other possibilities such as device aging or clogging of the valves from the solution itself?

6. During the implantation period, are rats allowed to move? if so, how to secure the catheters? Will the movement of the rats trigger the plunger due to inertial forces? For humans, will human motion large enough to trigger the plunger movement?

Novelty. In general, to be acceptable, a paper should represent an advance in understanding likely to influence thinking in the field. There should be a discernible reason why the work deserves the visibility of publication in a Nature journal rather than the best of the specialist journals.

Some sentences in the abstract should be revisited and novelty claim can be assessed appropriately, for example, in the Abstract: " but the lifetime of these devices is limited by the need for battery replacement and consequent replacement surgery." This is not true, Medtronic's insulin pump is located outside of the body and the battery can be replaced without surgery.

Reviewer #3 (Remarks to the Author)

The authors report a new batteryless, fully-implantable insulin pump that reproducibly delivers insulin in response to manual actuation by a static magnetic field. The proposed device is intended to replace implantable insulin infusion pumps for long-term pulsatile delivery of insulin, which overcomes the limitation of short battery life and the need for battery replacement. A refilling mechanism is proposed to combat depletion of the insulin reservoir. The implant design and actuation method are novel, however several significant challenges remain unanswered as elaborated below. Moreover, many experimental details were missing. Hence the manuscript is not ready for publication at this stage.

1) Stability of insulin for long-term use within the implant has not been demonstrated.

The manuscript only briefly indicates: "The stability and biological activity of insulin appeared to be maintained for up to 30 days under simulated biological conditions (Supplementary Fig. 3)."

In the caption of Suppl. Fig. 3, it says "An insulin solution (109 U/ml) in PBS (pH 7.4) was stored at 37 {degree sign}C for periods of 0, 10, 20 and 30 days and analyzed by RP-HPLC and circular dichroism (CD) spectroscopy." There is no place indicating where the insulin solution was stored. Was it stored in a glass vial or in the implant device?

The data presented in the manuscript are not convincing. The retention time of insulin measured by RP-HPLC is only indicative of aggregation or degradation of insulin. The CD spectra (Fig. 3b) at day 20 and day 30 showed some alteration of insulin structure. This result seems to be consistent with the in vivo data showing lower insulin concentration and less effect on glucose level after 11 days (Fig. 3).

Note that insulin inevitably denatures in the presence of hydrophobic surfaces, such as parylene C, given enough time at physiological conditions. The presence of denatured insulin induces rapid denaturation of normal insulin resulting in the formation of fibrils.

Denaturation/degradation/inactivation of insulin over time will decrease administered units of insulin per actuation. This can have serious consequences on the patient. A proper insulin stability study should be performed using the actual implant device and taking samples at various times for in vivo efficacy test, in addition to RP-HPLC and CD analysis.

2) Delivery of insulin on-demand has not been fully demonstrated.

Insulin delivery to the blood appears to decrease with time after day 11. After 11 days, the authors used consecutive two actuations to provide sufficient effect on glucose control. This practice is not suitable for clinical use. The currently used insulin infusion pumps have the mechanism for adjusting delivered amount either by patients or through closed-loop glucose sensor signal and controlling algorithms. To show the advantages of the proposed device over existing insulin delivery systems, the proposed system should have a mechanism to sense glucose concentration and adjust the amount of insulin to be delivered. In fact the patients would not know how many actuations are needed given unknown bioactivity of insulin in the implant.

3) The authors attribute the loss of insulin effect to development of a fibrous capsule, and attempt to compensate for this effect by doubling the number of actuations. This solution is not practical in the clinic. The authors should confirm the root cause for this discrepancy by measuring the drug dosing long after formation of a mature fibrous capsule. From Figure 3a, it appears that the delivered insulin dose is not stable after 11 days. Was there cellular infiltration and buildup of fibrous tissue in the outlet/check-valve orifice of the device after 30 days?

4) Refilling in the body is a potentially deadly scenario. Please comment on the reservoir pressure buildup and potential for insulin dumping during refilling procedure. The authors should examine the safety and reproducibility of such a procedure.

5) Please comment on potential interference from other magnetic sources and sources of mechanical failure over repeated uses. Have these issues been investigated?

6) Many experimental details were missing.

For example, it is unclear where insulin solution (109 U/ml) in PBS (pH 7.4) was stored at 37 {degree sign}C, as indicated above. Where was insulin purchased, what type? How was the insulin solution prepared?

How many rats in one treatment group? Were the rats fed or fast when they were given the insulin?

Reviewers' comments:

Reviewer #1 (Remarks to the Author):

The paper by Lee et al presents an interesting concept to use magnets in an implantable insulin pump to avoid the requirement for batteries. Any device that can improve the quality of life of diabetics, e.g. avoid multiple daily injections, decrease the incidence of hyper-/hypo-glycaemia, is welcome. The manuscript is clearly written. I have a few questions for the authors.

Major questions/comments:

1. Accurate activation and deactivation of the delivery mechanism is key. What is the maximum distance between the external and internal magnets before the mechanism stops working optimally? Are there any mechanisms to mitigate against accidental activation by a magnet in the vicinity?

As the reviewer pointed out, we have performed an additional experiment to assess the actuation of the MDP herein according to the distance between the external device and MDP. Our result showed that the MDP could be actuated at distances of up to 3 mm and in this range, the MDP could infuse the same amount of insulin. Considering the skin thickness (human: 2.15 ± 0.42 mm, rat: 1.18 ± 0.04 mm), therefore, our device is expected to properly function when implanted in a subcutaneous space.

To be actuated with the skin gap, our device needs an external magnet with at least 3000 G as the plunger was fixed in position due to the attraction between the plunger and barrel magnets (M_P and M_B , respectively). According to our literature survey, the largest magnetic field available in a regular life style was reported to be at most 2 G (e.g., a hair dryer). Therefore, a chance for an accidental activation of our device by the vicinity of the magnetic field is expected to be not high.

We have revised the manuscript and supplementary information accordingly.

Manuscript:

Page 6, line 1,

In vitro performance test. To investigate the performance of the MDP, an *in vitro* drug release test was performed, **where a gap of 1 mm was prepared between the external device and MDP to simulate the presence of the skin after implantation^{21,22}. In this work, the MDP could be actuated with gaps of up to 3 mm with the external magnet (M_E) of 3000 G (Supplementary Table 1).**

[Ref.]

21. Gibney, M.A., Arce, C.H., Byron, K.J. & Hirsch, L.J. Skin and subcutaneous adipose layer thickness in adults with diabetes at sites used for insulin injections: implications for needle length recommendations. *Curr. Med.Res.Opin.* **26**, 1519-1530 (2010).
22. Takeuchi, H., *et al.* Influence of skin thickness on the in vitro permeabilities of drugs through Sprague-Dawley rat or Yucatan micropig skin. *Biol. Pharm.Bull.* **35**, 192-202 (2012).

Supplementary information:

Supplementary Table 1 Actuation ability of the MDP according to the distance between the external device and MDP. The MDP could be actuated at distances of up to 3 mm and in this range, the MDP could infuse the same amount of insulin, as observed in our *in vitro* performance test. To be actuated, the MDP herein needs an external magnet with 3000 G. Therefore, considering the range of a magnetic field available in a regular life style (< 2 G)³, a chance for an accidental activation of the MDP is expected to be not high.

Gap between the external device and MDP (mm)	Actuation ability*
0	Y
1	Y
1.5	Y
2	Y
2.5	Y
3	Y
3.5	N

* Y : actuated; N : not actuated

[Supplementary Ref.]

- Hamdan, H. Measurements of ELF Electromagnetic Fields in Jordan Exposure Limits and Recommendations. *Dirasat: Eng. Sci.* **39** (2014).

2. Which insulin preparation was used in the test? It would be useful for application purpose to test different insulin preparations to check efficacy using the device as diabetics use different preparations.

In this work, we used an aqueous solution of a short acting insulin. As the reviewer suggested, we have performed the additional *in vivo* experiments with the MDPs filled with two different types of the insulin formulations, i.e., rapid acting and long acting insulins, respectively. We infused the similar doses of insulin for all experimental groups (rapid acting: 0.74 U; short acting: 0.8 U; long acting 0.74 U). As shown in the figure below, for all formulations, we observed the decrease in blood glucose level after actuation. As expected, the profiles of blood glucose level were different depending on the types of insulin used. For rapid acting insulin, the blood glucose level decreased and increased back more rapidly. For long acting insulin, the blood glucose level dropped relatively slowly but this lowered level was maintained for a longer period. The result, therefore, suggested that the MDP herein be applicable for the different types of insulin formulations.

We have revised the manuscript and supplementary information accordingly.

Therefore, the MDP herein could be applicable for a variety of insulin formulations and regimens to achieve a profile of blood glucose level needed for specific diabetic treatment (Supplementary Figs. 10 and 11).

Figure 1 Descriptive images of the MDP. (a) 3D schematic of the MDP. The MDP is composed of two distinct units: a drug reservoir and an actuator. The actuator is composed of a plunger and barrel. The drug reservoir is filled with insulin solution (109 U/ml) after assembly. To prepare the insulin solution, insulin in powder form (short acting; Sigma-Aldrich, MO, USA) was dissolved in sterile PBS at pH 7.4.

Supplementary information:

Supplementary Figure 10 Profiles of blood glucose level obtained with the three different insulin formulations. In addition to a short acting insulin mainly used in this study (i.e., MDP_1A (short acting insulin)), we additionally prepared the MDPs filled with a rapid acting and long acting insulin formulation (NovoRapid¹ and Lantus², respectively) to give the animal groups of MDP_1A (rapid acting insulin) and MDP_1A (long acting insulin), respectively. At 1 day after the MDP was implanted in STZ-induced diabetic rats, the blood glucose level was obtained at -1 to 720 min after insulin administration by a single actuation of the MDP. A similar dose of insulin was administered for all experimental groups (rapid acting: 0.74 U; short acting: 0.80 U; and long acting 0.74 U). For each animal group, four rats were employed for statistics. For all formulations, the decrease in blood glucose level was apparent after actuation and as expected, a specific profile of blood glucose level was observed for each type of the insulin formulations. For rapid acting insulin, the blood glucose level decreased and increased back more rapidly. For long acting insulin, the blood glucose level dropped relatively slowly and this lowered level was maintained for a longer period.

[Supplementary Ref.]

1. Reynolds, N.A. & Wagstaff, A.J. Insulin Aspart. *Drugs* **64**, 1957-1974 (2004).

2. Wang, F., Carabino, J.M. & Vergara, C.M. Insulin glargine: a systematic review of a long-acting insulin analogue. *Clin. Ther.* **25**, 1541-1577 (2003).

3. What happens if there is insulin solution leftover in the reservoir that is degraded and therefore no longer efficacious? Can this be easily removed following implantation? The authors mention using 1.2 ml of insulin in the reservoir when the reservoir can hold 3 ml, i.e. theoretically ~400 actuations, what is the rationale for this?

As the reviewer pointed out, we designed the MDP, abled to be replenished with a fresh insulin formulation. Considering the possible degradation of the leftover insulin, we have performed an additional experiment, where, during a replenishing procedure, we fully withdrew the insulin solution in the MDP and refilled it with the initial volume (1.2 ml) of a fresh one. In this experiment, a replenishing procedure was performed at 31 days while the MDP was still implanted in rats. After that, the profiles of plasma insulin concentration and blood glucose level were monitored for a longer time until 60 days. As shown in our revised Figure 3, the profiles of plasma insulin concentration and blood glucose level were reproducible to a large extent until 60 days.

In this work, we pursued to examine the feasibility of the MDP herein. For this, we prepared the MDP as a prototype, so that it could be small enough to be implantable in rats. Therefore, envisioning implantation in human, a larger MDP could be prepared to allow for a larger drug reservoir, which we believe can be designed more efficiently as unlike the conventional implantable drug-infusion pumps, the MDP herein does not need a space for battery.

We have revised the manuscript and supplementary information accordingly.

Manuscript:

Page 10, Line 5,

.....subcutaneous implantation of the MDP. **In this work, to examine the feasibility, we prepared the MDP as a prototype, small enough to be implantable in STZ-induced diabetic rats. Therefore, envisioning implantation in human, the MDP could be prepared to possess a larger volume, hence allowing for a larger drug reservoir. This could be designed more efficiently as the MDP herein does not need a space for battery.**

Figure 3. Profiles of (a,c) plasma insulin concentration and (b,d) blood glucose level with the four different animal groups: i) control group–diabetic rats with no treatment (n = 4), ii) S.C. injection group–diabetic rats subcutaneously injected with an insulin solution (0.8 U insulin) with a Hamilton microliter syringe at each of the scheduled times (n = 4), iii) MDP_1A group–diabetic rats implanted with the MDP in the subcutaneous space and treated with a single actuation (0.8 U insulin) at each of the scheduled times (n = 4); and iv) MDP_1A/2A group–diabetic rats implanted with the MDP and treated with a single actuation (0.8 U insulin) at each of the scheduled times until 11 days and two consecutive actuations (1.6 U insulin) at each of the scheduled times after 11 days (n = 4). For each actuation, the external device with M_E was applied and removed to the skin immediately above the implanted MDP. After insulin was delivered via actuation or injection, blood was withdrawn at $T_{\max, \text{insulin}}=60$ min to measure the maximum insulin concentration and also at $T_{\max, \text{glucose}}=120$ min to measure the minimum glucose level and its maximum decrease (Supplementary Fig. 5). (a,b) The plasma was sampled at scheduled times of 1, 2, 4, 7, 11, 16, 22 and 30 days with the four different animal groups. (c,d) After 30 days, we continued the experiment with the two different animals groups: i) control and iv) MDP_1A/2A groups. At 31 days, for the MDP_1A/2A group, we fully withdrew the insulin solution in the drug reservoir and refilled it with 1.2 ml of a fresh one while the MDP was still implanted (Supplementary Fig. 8). The plasma was sampled at scheduled times of 31, 40, 50 and 60 days.

4. How is the actuation volume controlled? By the amount of time the magnet is applied to the device? How envisage this applies to human use- calculations may be a bit too much for a patient?

In this work, to actuate the MDP, we applied the external device with M_E within 1 s. To examine the effect of the amount of time the magnet is applied, we have performed an additional experiment, where we have varied the application periods of M_E to 5 s and 10 s. As shown in a newly created Figure 9 in the Supplementary Information, we found that the delivered doses of insulin per actuation were quite similar regardless of the period for the external device application.

We have revised the manuscript and supplementary information accordingly.

Manuscript:

Page 10, Line 21,

Another advantage of our MDP is the relatively fast infusion of insulin, where **the delivered doses of insulin per actuation were quite similar regardless of the period for the external device application (Supplementary Fig. 9).**

Supplementary Information:

Supplementary Figure 9 Reproducibility assessment of the MDP with varied periods for the external device (M_E) application. Under the *in vitro* drug release experimental condition depicted in Fig. 2, the external device was applied to and removed from the MDP during the periods of < 1 s, 5 s and 10 s, respectively (i.e., MDP (< 1 s), MDP (5 s) and MDP (10 s), respectively). The results revealed that the delivered doses of insulin per actuation were quite similar regardless of the period for the external device application. Three distinct MDPs were tested for each period for the external device application.

5. Experimental details of the *in vitro* test were not given so it is difficult to visualize the experiment- e.g. how is distance of 4.5 mm from top of MDP to external magnet achieved? The authors state that there was almost no release of insulin in the period of no actuation; it would be useful to show the numerical data in the text. The error bars in Figure 2a, b are hard to see; how many devices were tested, how many times was this test repeated? Summary data showing average release with and without actuation would be useful, together with the variation as a separate figure.

We apologize for unclear presentation. We have modified the text and figures as suggested. For no actuation data, insulin was not detected as the concentration was below the LOQ (limit of quantification) with the measurement method employed herein. To show the error bars better, we have added the figures in the Supplementary Information, plotting a newly-released insulin profile at each of the actuations.

Manuscript:

Page 6, Line 4,

As shown in Figure 2a (**Supplementary Fig. 3(a)**), the release amount per actuation, 0.81 ± 0.04 U per actuation, was highly reproducible even with repeating actuations.

As shown in Figure 2b (Supplementary Fig. 3(b)), insulin was released only at the times of actuation, and no insulin was detected during the periods of non-actuation. Considering the LOQ (limit of quantification: 0.002 U/ml) with the measurement in this work, this result implied that even though there was a leak, it should be less than 0.02 U insulin during the longest non-actuation periods of 7 days ($< 3 \times 10^{-3}$ U per day).

Figure 2 *In vitro* insulin release profiles of the MDP. The MDP was fully immersed in pH 7.4 PBS and incubated in an incubator at 37°C while being continuously shaken at 50 rpm. At each actuation, the external magnet device was applied **at a constant distance from the top of the MDP by placing a 1 mm-thick glass slide between the external magnet device and MDP to simulate the presence of tissue after implantation**, and removed almost instantaneously (< 1 s) while the MDP was fully submerged in the release medium. An aliquot of the release medium was sampled at scheduled times and analyzed by high-performance liquid chromatography. Details on this procedure are described in the Methods section. **(a)**Thirty consecutive actuations were applied at intervals of 10 min, and an aliquot was sampled after each of the actuations. **With five distinct MDPs tested herein, the released amount of insulin was highly reproducible and was 0.81 ± 0.04 U per actuation.** **(b)** The MDP was actuated at predetermined times of 1, 3, 6, 10, 15, 22 and 28 days while fully immersed in PBS for 28 days. Aliquots were sampled immediately before and after each of the actuations. **With four distinct MDPs tested herein,** there was almost no release of insulin during the period of no actuation. The released amount of insulin was also highly reproducible and was 0.80 ± 0.09 U per actuation.

Supplementary Information:

Supplementary Figure 3 Insulin release profiles of the MDP. (a) Figure 2(a) and (b) Figure 2(b) were replotted to show the amount of newly released insulin per actuation. The released amounts of insulin were highly reproducible and were (a) 0.81 ± 0.04 U and (b) 0.80 ± 0.09 U per actuation. When there was no actuation, insulin was not detected with the measurement method employed in this work.

6. In vivo tests were only performed for 30 days; in fig 3a MDP_1A (and possibly MDP_1A/2A) appear to be drifting downward (the drift also evident in 3b), and it would make sense to continue the experiment till this stabilises. Was the insulin in the reservoir continually replenished during this period and if so it would be useful to show glucose concentrations and plasma insulin content after replenishment over this time course? Please state the predetermined time of day of actuation; how was this time decided upon- feeding behavior? It is difficult to rationalize the protocol described in the in vivo experiment with the human situation where insulin is injected at meal times to normalize blood glucose. What happens if you need to actuate at different times of the day e.g. to mimic meal times?

As the reviewer suggested, we have additionally performed a long-term experiment with the MDP_1A/2A group. At 31 days after MDP implantation, we fully withdrew the insulin solution in the MDP and refilled it with the initial volume (1.2 ml) of a fresh one. After that, we monitored the profiles of plasma insulin concentration and blood glucose level until 60 days. As shown in our revised Fig. 3, the insulin concentration and decreased glucose levels from 31 days were similar to those observed during the earlier period of 16-30 days (Fig. 3(a,b)). This result implied a long-term applicability of the MDP herein that could infuse the insulin in a reproducible manner even after a replenishing procedure.

For a practical reason in experimental design, we simply chose the specific days of actuation in this work. During *in vivo* experiments, the animals were freely accessible to food and water. However, as

the reviewer pointed out, we also agree that *in vivo* experiments with the human situation are needed, especially for a short-acting insulin that we used in this study. Therefore, we have performed an additional experiment, where the MDP after implantation was actuated twice in a day with an interval, mimicking the one between meal times. With diabetic rats in this work, an interval of 12 h was employed. As shown in Supplementary Fig. 11, after the first actuation, a glucose level dropped and increased back to a high level as expected with diabetic rats. This was observed to be repeated after the second actuation at 12 h. This result implied that the MDP herein could also be applicable for daily multiple insulin deliveries needed after meal times.

We have modified the manuscript and supplementary information accordingly.

Manuscript:

Page 7, Line 22,

To examine a long-term efficacy, we continued the experiment with the MDP 1A/2A group until 60 days, where the insulin in the MDP was fully replenished at 31 days. As shown in Fig. 3(c, d), the insulin concentration and decreased glucose levels from 31 days ($743.9 \pm 10.9 \mu\text{U/ml}$ and $-323.8 \pm 19.9 \text{ mg/dl}$, respectively) were similar to those observed during the earlier period at 16-30 days (Fig. 3(a,b)). This result implied a long-term applicability of the MDP herein that could infuse the insulin in a reproducible manner even after a replenishing procedure.

Page 11, Line 3

Therefore, the MDP herein could be applicable for a variety of insulin formulations and regimens to achieve a profile of blood glucose level needed for specific diabetic treatment (Supplementary Figs. 10 and 11).

Page 13, Line 13,

.....procedure for MDP implantation are described in Supplementary Fig. 12. **During the experiments, the animals were freely accessible to food and water.** To measure the plasma insulin concentration....

Figure 3. Profiles of (a,c) plasma insulin concentration and (b,d) blood glucose level with the four different animal groups: i) control group–diabetic rats with no treatment (n = 4), ii) S.C. injection group–diabetic rats subcutaneously injected with an insulin solution (0.8 U insulin) with a Hamilton microliter syringe at each of the scheduled times (n = 4), iii) MDP_1A group–diabetic rats implanted with the MDP in the subcutaneous space and treated with a single actuation (0.8 U insulin) at each of the scheduled times (n = 4); and iv) MDP_1A/2A group–diabetic rats implanted with the MDP and treated with a single actuation (0.8 U insulin) at each of the scheduled times until 11 days and two consecutive actuations (1.6 U insulin) at each of the scheduled times after 11 days (n = 4). For each actuation, the external device with M_E was applied and removed to the skin immediately above the implanted MDP. After insulin was delivered via actuation or injection, blood was withdrawn at $T_{\max, \text{insulin}}=60$ min to measure the maximum insulin concentration and also at $T_{\max, \text{glucose}}=120$ min to measure the minimum glucose level and its maximum decrease (Supplementary Fig. 5). (a,b) The plasma was sampled at scheduled times of 1, 2, 4, 7, 11, 16, 22 and 30 days with the four different animal groups. (c,d) After 30 days, we continued the experiment with the two different animals groups: i) control and iv) MDP_1A/2A groups. At 31 days, for the MDP_1A/2A group, we fully withdrew the insulin solution in the drug reservoir and refilled it with 1.2 ml of a fresh one while the MDP was still implanted (Supplementary Fig. 8). The plasma was sampled at scheduled times of 31, 40, 50 and 60 days.

Supplementary Information:

Supplementary Figure 11 *In vivo* profiles of blood glucose level with multiple daily actuations of the MDP. To give an insight of insulin delivery after each of the meal times, the MDP was actuated twice with an interval of 12 h. After the first actuation, a glucose level dropped and increased back to a high level as expected with diabetic rats, which was observed to be repeated in a similar pattern after the second actuation (n=4).

7. The authors mention that there was potentially a gradual release from the fibrotic capsule which may affect plasma insulin content, leading to a modification of the protocol. The authors should assess how much insulin was present in the fibrotic capsule at day 11 and following the 1 and 2 actuation protocols so as to determine how much insulin can be stored in the fibrotic cap (essentially a second reservoir). Could this be the reason why the insulin content at day 16 and 30 is higher than SC and takes longer to return to baseline (Fig. 4a), which is mirrored in the blood glucose content in 4b? This is important due to the dangers of hypoglycaemia. It would be interesting to check fibrotic capsule size for longer than 30 days post implant to determine if the fibrotic capsule may get any thicker/larger and worsen the second reservoir effect. The authors should also perform more detailed analysis of the inflammatory response: staining for markers of inflammatory in the sections and measurement of inflammatory markers in plasma should be performed.

As the reviewer pointed out, to assess the insulin amount possibly left in the fibrotic capsule, we have

performed an additional experiment. For this, we biopsied the whole capsule tissue around the MDP at two different days after implantation, i.e., at 16 days and 60 days, respectively. At each biopsy day, the MDP was actuated twice consecutively (1.6 U insulin delivery) and after 360 min, the whole capsule including the MDP was biopsied. From each of the biopsied capsule, we extracted the MDP from the surrounding capsule tissue and fully immersed the capsule in 5 ml of pH 7.4 PBS at 37 °C for 6 h. Then, the supernatant was analyzed with HPLC, as described in the Methods, to measure the insulin amount. Five and four animals were employed to obtain the biopsied tissues at 16 and 60 days, respectively. Our results revealed that 0.032 ± 0.011 U and 0.030 ± 0.008 U insulin was measured at 16 and 60 days, respectively, suggesting that more than 97.5% of the total amount of delivered insulin was diffused out of the fibrotic capsule during the first 360 min after actuations. This implied that a larger $AUC_{PK, \text{insulin}}$ and $AUC_{PD, \text{glucose}}$ were ascribed to greater systemic insulin exposure with the MDP_1A/2A group, which mostly occurred within hours after the MDP actuations.

With the biopsied capsule tissues, we have also performed histological analyses to measure the capsule thickness. As shown in the revised Fig. 5b, the capsule thicknesses did not increase further but rather stabilized from 16 days. The capsule thicknesses were measured to be 242 ± 56.9 μm , 243 ± 56.9 μm and 246 ± 41.7 μm at 16, 30 and 60 days after implantation, respectively.

As the reviewer suggested, we have also performed an additional experiment to better assess the inflammatory response. For this, we have assessed the H&E-stained tissues additionally at 60 days. Also, we have assessed the images of the CD68-stained tissues at both 30 and 60 days. As shown in the revised Figure 5(a, b), the overall inflammatory and foreign body reactions were minimal for all tested tissue locations around the MDP.

We have also measured some inflammatory markers in plasma, such as IL-1b, IL-6 and TNF- α , at 30 and 60 days after MDP implantation. For all animals, elevation of inflammatory markers was not observed and their levels in plasma were not different from the ones with the animals without MDP implantation.

We have modified the manuscript and supplementary information accordingly.

Manuscript:

Page 8, Line 18

In the MDP_1A/2A group, a larger dose of insulin was administered into the subcutaneous space and appeared to slowly diffuse into the blood stream across the fibrotic capsule around the MDP, resulting in greater systemic insulin exposure. **This process appeared to be completed within hours after MDP actuations (Supplementary Table 2).**

Page 9, Line 1

To assess *in vivo* biocompatibility, biopsied tissue samples around the MDP were examined by hematoxylin and eosin (H&E) and CD68 staining. As shown in Figure 5(a, b), the overall inflammatory and foreign body reactions were minimal for all tested tissue locations around the MDP. Also, elevation of inflammatory markers in plasma, such as IL-1b, IL-6 and TNF- α , was not observed at 30 and 60 days after MDP implantation (Supplementary Table 3).

The capsule thickness was $242 \pm 56.9 \mu\text{m}$ after 16 days, similar to the thicknesses at 30 and 60 days ($243 \pm 56.9 \mu\text{m}$ and $246 \pm 41.7 \mu\text{m}$, respectively). No sign of clogging was observed with the valve in the MDP at 60 days (Supplementary Fig. 7).

a

b

Figure 5 Representative histological images of the tissues around the MDP. Two distinct locations in the tissue were observed: the tissues near the reservoir body surface and near the outlet of the MDP. The asterisk (*) indicates the location of the implanted MDP. **n = 4; 16 days, n = 5; 30 days and n = 4; 60 days.** **To evaluate biocompatibility, we assessed the degree of inflammatory response in (a) H&E-stained and (b) CD68-stained tissues near both the reservoir body surface and the outlet of the MDP at 30 and 60 days after implantation.** The scale bars are 100 μ m. (c) Formation of collagen was assessed in both H&E- and MT-stained tissues near the outlet of the MDP **at 16, 30 and 60 days** after implantation. To measure the capsule thickness, the thinnest region of the capsule was selected in each image of the H&E-stained samples, as indicated by the arrows. The scale bars are 1 mm.

Supplementary Information:

Supplementary Table 2 Insulin amount left in fibrotic capsules. To assess the insulin amount possibly left in the fibrotic capsule, we biopsied the whole capsule including the MDP at two different days after implantation, i.e., at 16 days and 60 days, respectively (n = 5; 16 days, n = 4; 60 days). At each day, the MDP was actuated twice consecutively (1.6 U insulin delivery) and after 360 min, the biopsy was performed. From each of the biopsied capsule, we extracted the MDP and the surrounding capsule tissue was fully immersed in 5 ml of pH 7.4 PBS at 37 °C for 6 h. Then, the supernatant was analyzed with HPLC, as described in the Methods, to measure the insulin amount. Our results revealed that more than 97.5% of the total amount of delivered insulin was diffused out from the fibrotic capsule during the first 360 min after actuations.

Day	Insulin amount in fibrotic capsule (U)
16	0.032 \pm 0.011
60	0.030 \pm 0.008

Supplementary Table 3 Inflammatory markers in plasma. Inflammatory markers in plasma, such as IL-1b, IL-6 and TNF- α , were measured at 30 and 60 days after MDP implantation (n = 3; 30 days, n = 3; 60 days). For all animals, elevation of inflammatory markers was not observed and their levels in plasma were not different from the ones with the control animal group (i.e., the animals without MDP implantation).

	IL-1b (pg/ml)	IL-6 (pg/ml)	TNF- α (pg/ml)
Day 30	68 \pm 10	53 \pm 4.0	82 \pm 11
Day 60	68 \pm 4.0	56 \pm 3.0	87 \pm 8.2
Control	71 \pm 8.6	60 \pm 3.2	94 \pm 3.0

8. The device releases insulin in pulses but it is not the same as the pulsatile release exhibited by islets and the manuscript needs to be carefully worded so the two phenomenon are not confounded.

We also agree with the reviewer that our device is not the same as the pulsatile release exhibited by the biological system, such as islets. Our device, therefore, closely mimics insulin delivery via established clinical modalities, such as with an insulin pen or needle-syringe injections.

We have discussed this accordingly and added the relevant references in the manuscript.

Manuscript:

Page 10, Line 20,

Although not the same as the biological system, such as islets³⁰, this rapid infusion more closely mimics insulin delivery via established clinical modalities, such as with an insulin pen or needle-syringe injections¹.

[Ref.]

30. Veisheh, O., Tang, B.C., Whitehead, K.A., Anderson, D.G. & Langer, R. Managing diabetes with nanomedicine: challenges and opportunities. *Nat. Rev. Drug Discov.* **14**, 45-57 (2015).

Minor comments:

9. It would help the readers if the size of the device is indicated in the figures (e.g. Fig 1).

As the reviewer suggested, we have added the size information of the MDP in Fig. 1.

Reviewer #2 (Remarks to the Author):

This paper presents a magnetically operated pump that is able to deliver insulin. Animal model was presented to show evidence of the working principle. Overall, this work, while provides value in engineering design and perhaps the field of bioengineering, is unlikely to inspire thinking for general audience of Nature Communications.

Below, please find suggestions to improve the manuscript.

Data & methodology

1. Line 128-129: "... As shown in Figure 2b, insulin was released only at the times of actuation, and no leakage was observed during the periods of non-actuation." The authors implies that the valve has zero leakage. Perhaps the leakage amount is undetectable?

We apologize for unclear presentation. For no actuation data, insulin was not detected as the concentration was below the LOQ (limit of quantification) with the measurement method employed herein. We have modified the text in the manuscript as suggested.

Manuscript:

Page 6, Line 8,

As shown in Figure 2b (Supplementary Fig. 3(b)), insulin was released only at the times of actuation, and no insulin was detected during the periods of non-actuation. Considering the

LOQ (limit of quantification: 0.002 U/ml) with the measurement in this work, this result implied that even though there was a leak, it should be less than 0.02 U insulin during the longest non-actuation periods of 7 days ($< 3 \times 10^{-3}$ U per day).

2. Line 137, what is "S.C. injection group"? subcutaneous?

We apologize for unclear presentation. We have modified the text in the manuscript to clarify this.

Page 6, Line 21,

As shown in Fig. 3a, the insulin concentrations between the MDP_1A and **S.C. injection (i.e., subcutaneous injection) groups** were similar until day 11.

3. Will catheter length affect the amount of insulin delivering under the same magnetic field? Perhaps to due friction loss, more pressure from the plunger is required to push the same amount of fluid out as the catheter length increases. How would the present design to accommodate for that?

To test the efficacy of the MDP designed in this work, we implanted a whole body of the MDP without a catheter in diabetic rats, where the MDP could deliver insulin in an on-demand manner, i.e., only at the times of the external device application from the outside skin. In discussion, we tried to think further and thus, considered a possible clinical issue, i.e., local lipohypertrophy at the site of subcutaneous insulin infusion. In this scenario, we believe that a catheter-connected MDP can be advantageous, where insulin can be infused via a catheter into the intraperitoneal space while the MDP is implanted subcutaneously for improved access to the external magnetic device.

In this specific scenario, we agree that the catheter length would be a factor influencing the amount of insulin release. Therefore, we have performed an additional experiment, where the amount of insulin release per actuation was measured with varied catheter lengths under the *in vitro* experimental condition, as depicted in Fig. 2. Although the amount of released insulin decreased with the catheter length, our result revealed that insulin could still be released by actuation in a reproducible manner with a catheter length of up to 15 cm and this was reported to be similar to an anatomical distance between the subcutaneous and intraperitoneal space in humans. This result indicated a reproducible volume of liquid infused per actuation and thus, the dose of insulin per actuation could be accommodated by employing a proper concentration of insulin formulation to be filled in the drug reservoir.

We have modified the manuscript and supplementary information accordingly.

Manuscript:

Page 11, Line 15

In this scenario, a catheter-connected MDP can be advantageous. For example, insulin can be infused via a catheter into the intraperitoneal space while implanting the MDP subcutaneously for improved access to the external magnetic device³⁸ (Supplementary Video 2 **and Table 4**).

Supplementary Information:

Supplementary Table 4 Amount of released insulin with varied catheter lengths. The MDP without a catheter (i.e., a catheter, 0 cm in length) and the ones connected with a catheter, 10 and 15 cm in length, respectively, were each actuated under the *in vitro* experimental condition, as depicted in Fig. 2. Although the amount of released insulin decreased with the catheter length, insulin could still be released in a reproducible manner with a catheter length of up to 15 cm and this was reported to be similar to an anatomical distance between the subcutaneous and intraperitoneal space in humans⁴. The result indicated a reproducible volume of liquid infused per actuation and thus, the dose of insulin could be accommodated by employing a proper concentration of insulin formulation to be filled in the drug reservoir.

Catheter length (cm)	Insulin release amount (U)
0	0.81 ± 0.04
10	0.74 ± 0.04
15	0.68 ± 0.04

[Supplementary Ref.]

4. Thompson, J.S. & Duckworth, W.C. Insulin pumps and glucose regulation. *World J. Surg.* **25**, 523-526 (2001).

4. Line 28:"...implantable insulin pump that can be actuated by a static magnetic field.". The statement is not true, the magnetic field has to vary, otherwise, one would see no motion from the plunger. So a user uses a magnet to move the plunger, the speed of the user moves the magnetic towards or away from the plunger will affect the speed of the plunger, and the speed of the fluid, will this speed differences affect insulin delivery?

In this work, to actuate the MDP, we applied the external device with M_E within 1 s. To examine the effect of the amount of time the magnet is applied, we have performed an additional experiment, where we have varied the application periods of M_E to 5 s and 10 s. As shown in a newly created figure in the Supplementary Information (Supplementary Fig. 9), we found that the delivered doses of insulin per actuation were quite similar regardless of the period for the external device application.

We agree that a magnetic field should vary to actuate the plunger, which occurs while the external device magnet gets close to and away from the MDP. Therefore, we have removed the word, static, from the text in the manuscript.

We have revised the manuscript and supplementary information accordingly.

Manuscript:

Page 10, Line 21,

Another advantage of our MDP is the relatively fast infusion of insulin, **where the delivered doses of insulin per actuation were quite similar regardless of the period for the external device application (Supplementary Fig. 9).**

Supplementary Information:

Supplementary Figure 9 Reproducibility assessment of the MDP with varied periods for the external device (M_E) application. Under the *in vitro* drug release experimental condition depicted in Fig. 2, the external device was applied to and removed from the MDP during the periods of < 1 s, 5 s and 10 s, respectively (i.e., MDP (< 1 s), MDP (5 s) and MDP (10 s), respectively). The results revealed that the delivered doses of insulin per actuation were quite similar regardless of the period for the external device application. Three distinct MDPs were tested for each period for the external device application.

5. As the implantation period is longer than 11 days, insulin delivery goes down, the authors argue that the formation of a fibrous capsule could be to blame. It would be easy to just look at the catheter and determine for sure whether or not fibrous tissues is indeed the cause. Can the authors rule out other possibilities such as device aging or clogging of the valves from the solution itself?

As stated above, we implanted a whole body of the MDP without a catheter in diabetic rats, where the MDP could deliver insulin in an on-demand manner, i.e., only at the times of the external device application from the outside skin. Therefore, instead of a catheter, we have examined the valve at 60 days after MDP implantation, which did not show any apparent sign of clogging.

We have also performed the *in vivo* experiment with the MDP_1A/2A group for a longer time until 60 days. As shown in our revised Fig. 3, the insulin concentration and decreased glucose levels from 31 days were similar to those observed during the earlier period of 16-30 days (Fig. 3(a,b)). This result implied a long-term applicability of the MDP herein that could infuse the insulin in a reproducible manner.

We have also performed histological analyses to measure the capsule thickness additionally at 60 days. As shown in the revised Fig. 5b, the capsule thicknesses did not increase further but rather stabilized from 16 days. The capsule thicknesses were measured to be $242 \pm 56.9 \mu\text{m}$, $243 \pm 56.9 \mu\text{m}$ and $246 \pm 41.7 \mu\text{m}$ at 16, 30 and 60 days after implantation, respectively.

We have modified the manuscript and supplementary information accordingly.

Manuscript:

Page 7, Line 22,

To examine a long-term efficacy, we continued the experiment with the MDP 1A/2A group until 60 days, where the insulin in the MDP was fully replenished at 31 days. As shown in Fig. 3(c, d), the insulin concentration and decreased glucose levels from 31 days ($743.9 \pm 10.9 \mu\text{U/ml}$ and $323.8 \pm 19.9 \text{ mg/dl}$, respectively) were similar to those observed during the earlier period at 16-30 days (Fig. 3(a,b)). This result implied a long-term applicability of the MDP herein that could infuse the insulin in a reproducible manner even after a replenishing procedure.

Page 9, Line 11

The capsule thickness was $242 \pm 56.9 \mu\text{m}$ after 16 days, similar to the thicknesses at 30 and 60 days ($243 \pm 56.9 \mu\text{m}$ and $246 \pm 41.7 \mu\text{m}$, respectively). No sign of clogging was observed with the valve in the MDP at 60 days (Supplementary Fig. 7).

Page 25

Figure 3. Profiles of (a,c) plasma insulin concentration and (b,d) blood glucose level with the four different animal groups: i) control group–diabetic rats with no treatment ($n = 4$), ii) S.C. injection group–diabetic rats subcutaneously injected with an insulin solution (**0.8 U insulin**) with a Hamilton microliter syringe at each of the scheduled times ($n = 4$), iii) MDP_1A group–diabetic rats implanted with the MDP in the subcutaneous space and treated with a single actuation (**0.8 U insulin**) at each of the scheduled times ($n = 4$); and iv) MDP_1A/2A group–diabetic rats implanted with the MDP and treated with a single actuation (**0.8 U insulin**) at each of the scheduled times until 11 days and two consecutive actuations (**1.6 U insulin**) at each of the scheduled times after 11 days ($n = 4$). For each actuation, the external device with M_E was applied and removed to the skin immediately above the implanted MDP. After insulin was delivered via actuation or injection, blood was withdrawn at $T_{\max, \text{insulin}}=60$ min to measure the maximum insulin concentration and also at $T_{\max, \text{glucose}}=120$ min to measure the minimum glucose level and its maximum decrease (Supplementary Fig. 5). **(a,b) The plasma was sampled at scheduled times of 1, 2, 4, 7, 11, 16, 22 and 30 days with the four different animal groups. (c,d) After 30 days, we continued the experiment with the two different animals groups: i) control and iv) MDP_1A/2A groups. At 31 days, for the MDP_1A/2A group, we fully withdrew the insulin solution in the drug reservoir and refilled it with 1.2 ml of a fresh one while the MDP was still implanted (Supplementary Fig. 8). The plasma was sampled at scheduled times of 31, 40, 50 and 60 days.**

Figure 5 Representative histological images of the tissues around the MDP. Two distinct locations in the tissue were observed: the tissues near the reservoir body surface and near the outlet of the MDP. The asterisk (*) indicates the location of the implanted MDP. **n = 4; 16 days, n = 5; 30 days and n = 4; 60 days. To evaluate biocompatibility, we assessed the degree of inflammatory response in (a) H&E-stained and (b) CD68-stained tissues near both the reservoir body surface and the outlet of the MDP at 30 and 60 days after implantation.** The scale bars are 100 μ m. (c) Formation of collagen was assessed in both H&E- and MT-stained tissues near the outlet of the MDP **at 16, 30 and 60 days** after implantation. To measure the capsule thickness, the thinnest region of the capsule was selected in each image of the H&E-stained samples, as indicated by the arrows. The scale bars are 1 mm.

Supplementary Information:

Supplementary Figure 7 Representative images of the intact valve and the one extracted from the MDP biopsied at 60 days after implantation. No sign of clogging was seen with the valve in the implanted MDP.

6. During the implantation period, are rats allowed to move? if so, how to secure the catheters? Will the movement of the rats trigger the plunger due to inertial forces? For humans, will human motion large enough to trigger the plunger movement?

As stated above, we implanted a whole body of the MDP without a catheter in diabetic rats, where the MDP could deliver insulin in an on-demand manner, i.e., only at the times of the external device application from the outside skin. Therefore, in the scope of this study, we did not consider securing the catheter after implantation.

In an aspect of an inertial force, the attraction force between the plunger and barrel magnets (M_P and M_B , respectively) was measured to be 12.5 N, which could fix their position to not release insulin during the period of no actuation. With the mass of the plunger used in this study (9.8×10^{-4} kg), the acceleration needed to overcome this attraction force and move the plunger was calculated to be 1.28×10^4 m/s², which was a more than 1000 times larger than gravitational acceleration (9.8 m/s²). In this sense, a chance for an accidental movement of the plunger in the MDP herein is expected to be very low.

We have modified the manuscript and supplementary information accordingly.

Manuscript:

Page 5, Line 7

Magnets were placed in the heads of the barrel and plunger (i.e., M_B and M_P , respectively) to face each other with opposite polarities and therefore attach to each other **strongly enough to prevent accidental movement of the plunger (also see ‘Theoretical assessment of magnetic forces’ in**

Supplementary Information).

Supplementary information:

Page 30

Theoretical assessment of magnetic forces between the plunger and barrel magnets. The attraction force between the plunger and barrel magnets (M_p and M_B , respectively) was measured to be 12.5 N (Advanced force measurement 9830, Interface, USA), which could fix their position to not release insulin during the period of no actuation. With the mass of the plunger used in this study (9.8×10^{-4} kg), therefore, the acceleration needed to overcome this attraction force and move the plunger was calculated to be $1.28 \times 10^4 \text{ m/s}^2$, which was more than 1000 times larger than gravitational acceleration (9.8 m/s^2). In this sense, a chance for an accidental movement of the plunger in the MDP herein is expected to be very low.

Novelty. In general, to be acceptable, a paper should represent an advance in understanding likely to influence thinking in the field. There should be a discernible reason why the work deserves the visibility of publication in a Nature journal rather than the best of the specialist journals.

Some sentences in the abstract should be revisited and novelty claim can be assessed appropriately, for example, in the Abstract: " but the lifetime of these devices is limited by the need for battery replacement and consequent replacement surgery." This is not true, Medtronic's insulin pump is located outside of the body and the battery can be replaced without surgery.

In this work, we demonstrated a fully-implantable infusion pump enabled with on-demand, pulsatile release of insulin. Most importantly, the pump herein can be actuated to infuse insulin via an externally applied magnetic field, thus enabling non-invasive, on-demand insulin delivery after one-time implantation. Therefore, the pump can be operated without a battery. We also designed the pump to be refilled, allowing for potentially semi-permanent use. The desired dose of insulin can be accurately and reproducibly controlled by varying the number of actuations, resulting in pharmacokinetics and pharmacodynamics profiles similar to those of conventional subcutaneous insulin injections. This control can be achieved simply by varying the number of applications of a pen-type magnet on the outside skin above the implanted insulin-infusion pump. In this sense, to our knowledge, this study is the first to describe a simple-assembly, fully-implantable pump for batteryless, on-demand insulin infusion via a magnetic field.

Therefore, we believe that our pump in this study is different from a Medtronic insulin pump that is designed to be located outside of the body. For a fully implantable pump, surgery is mostly inevitable to replace battery. However, our pump can be actuated via a magnetic field, hence no need of battery.

To clarify that our pump is designed to be fully implantable and thus, it is different from the Medtronic pump, we have modified the text in the manuscript.

Manuscript:

Page 2, Line 4,

Here, we propose a batteryless, **fully-implantable** insulin pump that can be actuated by a magnetic

field.

Page 3, Line 7,

However, MDIs are often painful and erroneous, and CSII requires a bulky, external device⁴.

[Ref.]

4. Eugster, E.A., *et al.* Position statement: continuous subcutaneous insulin infusion in very young children with type 1 diabetes. *Pediatrics* **118**, 1244-1249 (2006).

Page 4, Line 2,

To our knowledge, this study is the first to describe a simple-assembly, fully-implantable pump for batteryless, on-demand insulin infusion via a magnetic field.

Page 9, Line 18,

In this work, we designed a fully-implantable MDP for semi-permanent use.

Reviewer #3 (Remarks to the Author):

The authors report a new batteryless, fully-implantable insulin pump that reproducibly delivers insulin in response to manual actuation by a static magnetic field. The proposed device is intended to replace implantable insulin infusion pumps for long-term pulsatile delivery of insulin, which overcomes the limitation of short battery life and the need for battery replacement. A refilling mechanism is proposed to combat depletion of the insulin reservoir. The implant design and actuation method are novel, however several significant challenges remain unanswered as elaborated below. Moreover, many experimental details were missing. Hence the manuscript is not ready for publication at this stage.

1. Stability of insulin for long-term use within the implant has not been demonstrated.

The manuscript only briefly indicates: "The stability and biological activity of insulin appeared to be maintained for up to 30 days under simulated biological conditions (Supplementary Fig. 3)." In the caption of Suppl. Fig. 3, it says "An insulin solution (109 U/ml) in PBS (pH 7.4) was stored at 37 °C for periods of 0, 10, 20 and 30 days and analyzed by RP-HPLC and circular dichroism (CD) spectroscopy." There is no place indicating where the insulin solution was stored. Was it stored in a glass vial or in the implant device? The data presented in the manuscript are not convincing. The retention time of insulin measured by RP-HPLC is only indicative of aggregation or degradation of insulin. The CD spectra (Fig. 3b) at day 20 and day 30 showed some alteration of insulin structure. This result seems to be consistent with the in vivo data showing lower insulin concentration and less effect on glucose level after 11 days (Fig. 3).

Note that insulin inevitably denatures in the presence of hydrophobic surfaces, such as parylene C, given enough time at physiological conditions. The presence of denatured insulin induces rapid denaturation of normal insulin resulting in the formation of fibrils. Denaturation/degradation/inactivation of insulin over time will decrease administered units of insulin per actuation. This can have serious consequences on the patient. A proper insulin stability study should be performed using the actual implant device and taking samples at

various times for *in vivo* efficacy test, in addition to RP-HPLC and CD analysis.

We apologize for unclear presentation. We performed the stability test while the insulin solution was stored in the MDP prepared in this work. At scheduled times of 0, 10, 20 and 30 days, we extracted the insulin solution from the MDP for analysis. As the reviewer pointed out, the data from the HPLC analysis would suggest no aggregation or degradation of insulin. The CD spectra were similar between testing periods to a large extent, implying that for most of insulin in the MDP, the secondary structure remained unchanged after 30 days of incubation at body temperature.

As the reviewer pointed out, we have performed an additional *in vivo* experiment to further test the efficacy and stability of insulin after incubation. For this, a fresh insulin solution and a solution extracted from the MDP after incubation for 30 days were subcutaneously injected to diabetic rats, respectively. For this, a fresh insulin solution was prepared at the same concentration (109 U/ml) as the one initially used for stability test and the same volume (7.4 μ l) of each of the solutions was injected. As shown in our newly created Supplementary Fig. 4c, the profiles of blood glucose level were similar between two groups to a large extent, implying that most of insulin stored in the MDP for 30 days could still effectively lower the blood glucose level.

We have modified the supplementary information accordingly.

Supplementary Information:

Supplementary Figure 4 Stability evaluation of insulin. An insulin solution (109 U/ml) in PBS (pH 7.4) was stored in the MDP at 37 °C for periods of 0, 10, 20 and 30 days and analyzed by RP-HPLC and circular dichroism (CD) spectroscopy. (a) The RP-HPLC analysis was performed following the same procedure described in Figure 2 in the main text. The uniform intensity and constant retention time of insulin among the tested incubation periods indicate that the insulin stored in the MDP was retained without aggregation or degradation until day 30 of incubation at body temperature. (b) CD spectra were obtained using a CD spectrometer (J-810, Jasco, Japan). A quartz cuvette with a 1-mm

path length was used, and spectra were scanned at 200–250 nm. **The CD spectra were seen to be similar among the tested incubation periods to a large extent, implying that for most of insulin in the MDP, the secondary structure remained unchanged after 30 days of incubation at body temperature. (c) A fresh insulin solution and a solution extracted from the MDP after incubation for 30 days were subcutaneously injected to diabetic rats, respectively (n = 4; fresh insulin, n = 4; insulin stored in the MDP for 30 days). A fresh insulin solution was prepared at the same concentration (109 U/ml) as the one initially used for stability test with the MDP and the same volume (7.4 µl) of each of the solutions was injected. Between two groups, the profiles of blood glucose level were similar, implying that most of insulin stored in the MDP for 30 days could still effectively lower the blood glucose level.**

2. Delivery of insulin on-demand has not been fully demonstrated.

Insulin delivery to the blood appears to decrease with time after day 11. After 11 days, the authors used consecutive two actuations to provide sufficient effect on glucose control. This practice is not suitable for clinical use. The currently used insulin infusion pumps have the mechanism for adjusting delivered amount either by patients or through closed-loop glucose sensor signal and controlling algorithms. To show the advantages of the proposed device over existing insulin delivery systems, the proposed system should have a mechanism to sense glucose concentration and adjust the amount of insulin to be delivered. In fact the patients would not know how many actuations are needed given unknown bioactivity of insulin in the implant.

The currently available devices for insulin delivery, such as the insulin pumps and pens, are designed to control the infusion volume of liquid and by this, the delivered dose of insulin can be adjusted with a known concentration of insulin solution. Our MDP can also infuse a specific volume of liquid per actuation and thus, the delivered amount of insulin can be adjusted by the number of actuations with a known concentration of insulin solution in the MDP. In a way to show this advantage, we increased the number of actuations after 11 days to compensate the decreased insulin efficacy caused by fibrotic capsule formation. However, as fibrotic capsule formation around the nondegradable implant like the MPD herein is natural and inevitable, we can also envision delaying the operation of the MDP until formation of the fibrous capsule is complete.

In this work, we focused more on the batteryless operation of our fully implantable MDP. Thus, we demonstrated that our MDP could be actuated to infuse insulin via an externally applied magnetic field, thus enabling non-invasive insulin delivery after one-time implantation. We thought that our strategy could replace the established clinical modalities for insulin delivery, such as an insulin pen or needle-syringe injection.

As the reviewer pointed out, we also agree that our MDP can be more advantageous with a closed-loop glucose control.

We have modified the text in the manuscript accordingly.

Manuscript:

Page 10, Line 24,

Although not the same as the biological system, such as islets³⁰, this rapid infusion more closely mimics insulin delivery via the established clinical modalities, such as with an insulin pen or

needle-syringe injections¹.

Page 11, Line 6,

Dose adjustment by varying the number of consecutive external device applications can also be easily performed because of the fast mechanical response between the plunger and barrel. At each actuation, the MDP infuses a specific volume of liquid, and therefore the dose of each actuation can also be varied by changing the concentration of insulin solution stored in the drug reservoir. Thus, we envision that a dose of prandial insulin can be customized within the typically prescribed range of 0.5–1.0 U, depending on the patient's needs for precise insulin dosing³². **This dose adjustment would be more effective and convenient when combined with a closed-loop glucose sensor^{33,34}.**

[Ref.]

33. Bergenstal, R.M., *et al.* Effectiveness of sensor-augmented insulin-pump therapy in type 1 diabetes. *N. Engl. J. Med.* **363**, 311-320 (2010).
34. El-Khatib, F.H., Russell, S.J., Nathan, D.M., Sutherlin, R.G. & Damiano, E.R. A bihormonal closed-loop artificial pancreas for type 1 diabetes. *Sci. Transl. Med.* **2**, 27ra27-27ra27 (2010).

Page 11, Line 17,

Fibrotic capsule formation occurs gradually over weeks³⁶, and thus to ensure reproducible systemic drug exposure, we envision delaying the operation of the MDP herein until formation of the fibrous capsule is complete¹⁴.

[Ref.]

36. Park, S., *et al.* Acute suppression of TGF- β with local, sustained release of tranilast against the formation of fibrous capsules around silicone implants. *J. Control. Release* **200**, 125-137 (2015).
14. Farra, R., *et al.* First-in-human testing of a wirelessly controlled drug delivery microchip. *Sci. Transl. Med.* **4**, 122ra121-122ra121 (2012).

3. The authors attribute the loss of insulin effect to development of a fibrous capsule, and attempt to compensate for this effect by doubling the number of actuations. This solution is not practical in the clinic. The authors should confirm the root cause for this discrepancy by measuring the drug dosing long after formation of a mature fibrous capsule. From Figure 3a, it appears that the delivered insulin dose is not stable after 11 days. Was there cellular infiltration and buildup of fibrous tissue in the outlet/check-valve orifice of the device after 30 days?

We have performed histological analyses to measure the capsule thickness additionally at 60 days after MDP implantation. As shown in the revised Fig. 5b, the capsule thicknesses did not increase further but rather stabilized from 16 days. The capsule thicknesses were measured to be 242 ± 56.9 μm , 243 ± 56.9 μm and 246 ± 41.7 μm at 16, 30 and 60 days after implantation, respectively.

We have also examined the valve extracted from the MDP biopsied at 60 days after implantation, which did not show any apparent sign of clogging.

We have also performed the *in vivo* experiment with the MDP_1A/2A group for a longer time until 60

days. As shown in our revised Fig. 3, the insulin concentration and decreased glucose levels from 31 days were similar to those observed during the earlier period of 16-30 days (Fig. 3(a,b)). This result implied a long-term applicability of the MDP herein that could infuse the insulin in a reproducible manner.

We have modified the manuscript and supplementary information accordingly.

Manuscript:

Page 7, Line 22,

To examine a long-term efficacy, we continued the experiment with the MDP 1A/2A group until 60 days, where the insulin in the MDP was fully replenished at 31 days. As shown in Fig. 3(c, d), the insulin concentration and decreased glucose levels from 31 days ($743.9 \pm 10.9 \mu\text{U/ml}$ and $-323.8 \pm 19.9 \text{ mg/dl}$, respectively) were similar to those observed during the earlier period at 16-30 days (Fig. 3(a,b)). This result implied a long-term applicability of the MDP herein that could infuse the insulin in a reproducible manner even after a replenishing procedure.

Page 9, Line 11

The capsule thickness was $242 \pm 56.9 \mu\text{m}$ after 16 days, similar to the thicknesses at 30 and 60 days ($243 \pm 56.9 \mu\text{m}$ and $246 \pm 41.7 \mu\text{m}$, respectively). No sign of clogging was observed with the valve in the MDP at 60 days (Supplementary Fig. 7).

Page 25

Figure 3. Profiles of (a,c) plasma insulin concentration and (b,d) blood glucose level with the

four different animal groups: i) control group–diabetic rats with no treatment (**n = 4**), ii) S.C. injection group–diabetic rats subcutaneously injected with an insulin solution (**0.8 U insulin**) with a Hamilton microliter syringe at each of the scheduled times (**n = 4**), iii) MDP_1A group–diabetic rats implanted with the MDP in the subcutaneous space and treated with a single actuation (**0.8 U insulin**) at each of the scheduled times (**n = 4**); and iv) MDP_1A/2A group–diabetic rats implanted with the MDP and treated with a single actuation (**0.8 U insulin**) at each of the scheduled times until 11 days and two consecutive actuations (**1.6 U insulin**) at each of the scheduled times after 11 days (**n = 4**). For each actuation, the external device with M_E was applied and removed to the skin immediately above the implanted MDP. After insulin was delivered via actuation or injection, blood was withdrawn at $T_{\max, \text{insulin}}=60$ min to measure the maximum insulin concentration and also at $T_{\max, \text{glucose}}=120$ min to measure the minimum glucose level and its maximum decrease (Supplementary Fig. 5). **(a,b) The plasma was sampled at scheduled times of 1, 2, 4, 7, 11, 16, 22 and 30 days with the four different animal groups. (c,d) After 30 days, we continued the experiment with the two different animals groups: i) control and iv) MDP 1A/2A groups. At 31 days, for the MDP 1A/2A group, we fully withdrew the insulin solution in the drug reservoir and refilled it with 1.2 ml of a fresh one while the MDP was still implanted (Supplementary Fig. 8). The plasma was sampled at scheduled times of 31, 40, 50 and 60 days.**

a

Figure 5 Representative histological images of the tissues around the MDP. Two distinct locations in the tissue were observed: the tissues near the reservoir body surface and near the outlet of the MDP. The asterisk (*) indicates the location of the implanted MDP. **n = 4; 16 days, n = 5; 30 days and n = 4; 60 days.** **To evaluate biocompatibility, we assessed the degree of inflammatory response in (a) H&E-stained and (b) CD68-stained tissues near both the reservoir body surface and the outlet of the MDP at 30 and 60 days after implantation.** The scale bars are 100 μ m. (c) Formation of collagen was assessed in both H&E- and MT-stained tissues near the outlet of the MDP **at 16, 30 and 60 days** after implantation. To measure the capsule thickness, the thinnest region of the capsule was selected in each image of the H&E-stained samples, as indicated by the arrows. The scale bars are 1 mm.

Supplementary Information:

Supplementary Figure 7 Representative images of the intact valve and the one extracted from the MDP biopsied at 60 days after implantation. No sign of clogging was seen with the valve in the implanted MDP.

4. Refilling in the body is a potentially deadly scenario. Please comment on the reservoir pressure buildup and potential for insulin dumping during refilling procedure. The authors should examine the safety and reproducibility of such a procedure.

As the reviewer suggested, at 31 days after MDP implantation, we have fully withdrawn the insulin solution in the MDP and refilled it with the initial volume (1.2 ml) of a fresh one. This refilling procedure was performed while the MDP was still implanted. After that, we did not observe any sign of hypoglycemia possibly caused by insulin dumping with all tested animals. Due to the valve located at the outlet, a slight change in reservoir pressure did not appear to cause considerable insulin burst release.

We have modified the text in the manuscript accordingly.

Manuscript:

Page 8, Line 2,

After a replenishing procedure, we did not observe any sign of hypoglycemia with all tested animals. Due to the valve located at the outlet, a slight change in reservoir pressure did not appear to cause considerable insulin burst release.

5. Please comment on potential interference from other magnetic sources and sources of mechanical failure over repeated uses. Have these issues been investigated?

To be actuated, our device needs about an external magnet with at least 3000 G as the plunger was fixed in position due to the attraction between the plunger and barrel magnets (M_P and M_B , respectively). According to our literature survey, the largest magnetic field available in a regular life style was reported to be at most 2 G (e.g., a hair dryer). Therefore, a chance for an accidental activation of our device by the vicinity of the magnetic field is expected to be not high.

In our device, the plunger was not tightly inserted in the barrel but with a gap of about 150 μm . Therefore, with over repeated uses, the structural degradation due to friction is expected to be not high. As shown in Fig. 2(a), insulin was released reproducibly with 30 consecutive actuations at intervals of 10 min.

We have revised the manuscript and supplementary information accordingly.

Manuscript:

Page, 5, Line 17,

The plunger was not tightly inserted in the barrel (gap = 150 μm) to allow for smooth and reproducible actuations.

Page 6, line 1,

To investigate the performance of the MDP, an *in vitro* drug release test was performed, **where a gap of 1 mm was prepared between the external device and MDP to simulate the presence of the skin after implantation^{21,22}. In this work, the MDP could be actuated with this gap of up to 3 mm with the external magnet (M_E) of 3000 G (Supplementary Table 1).** As shown in Figure 2a,.....

Supplementary Information:

Supplementary Table 1 Actuation ability of the MDP according to the distance between the external device and MDP. The MDP could be actuated at distances of up to 3 mm and in this range, the MDP could infuse the same amount of insulin, as observed in our *in vitro* performance test. To be actuated, the MDP herein needs an external magnet with 3000 G. Therefore, considering the range of a magnetic field available in a regular life style ($< 2 \text{ G}$)³, a chance for an accidental activation of the MDP is expected to be not high.

Gap between the external device and MDP (mm)	Actuation ability*
0	Y
1	Y
1.5	Y
2	Y
2.5	Y
3	Y
3.5	N

Y : actuated; N : not actuated

[Supplementary Ref.]

- Hamdan, H. Measurements of ELF Electromagnetic Fields in Jordan Exposure Limits and Recommendations. *Dirasat: Eng. Sci.* **39** (2014).

6. Many experimental details were missing.

For example, it is unclear where insulin solution (109 U/ml) in PBS (pH 7.4) was stored at 37°C, as indicated above. Where was insulin purchased, what type? How was the insulin solution prepared?

We apologize for unclear presentation. We have modified the text to clarify the experimental details.

Manuscript:

Figure 1 Descriptive images of the MDP. (a) 3D schematic of the MDP. The MDP is composed of two distinct units: a drug reservoir and an actuator. The actuator is composed of a plunger and barrel. The drug reservoir is filled with insulin solution (109 U/ml) after assembly. To prepare the insulin solution, insulin in powder form (short acting; Sigma-Aldrich, MO, USA) was dissolved in sterile PBS at pH 7.4.

Figure 2 *In vitro* insulin release profiles of the MDP. The MDP was fully immersed in pH 7.4 PBS and incubated in an incubator at 37°C while being continuously shaken at 50 rpm. At each actuation, the external magnet device was applied at a constant distance from the top of the MDP by placing a 1 mm-thick glass slide between the external magnet device and MDP to simulate the presence of tissue after implantation, and removed almost instantaneously (< 1 s) while the MDP was fully submerged in the release medium. An aliquot of the release medium was sampled at scheduled times and analyzed by high-performance liquid chromatography. Details on this procedure are described in the Methods section. (a) Thirty consecutive actuations were applied at intervals of 10 min, and an aliquot was sampled after each of the actuations. With five distinct MDPs tested herein, the

released amount of insulin was highly reproducible and was 0.81 ± 0.04 U per actuation. (b) The MDP was actuated at predetermined times of 1, 3, 6, 10, 15, 22 and 28 days while fully immersed in PBS for 28 days. Aliquots were sampled immediately before and after each of the actuations. **With four distinct MDPs tested herein,** there was almost no release of insulin during the period of no actuation. The released amount of insulin was also highly reproducible and was 0.80 ± 0.09 U per actuation.

Figure 3. Profiles of (a,c) plasma insulin concentration and (b,d) blood glucose level with the four different animal groups: i) control group—diabetic rats with no treatment (**n = 4**), ii) S.C. injection group—diabetic rats subcutaneously injected with an insulin solution (**0.8 U insulin**) with a Hamilton microliter syringe at each of the scheduled times (**n = 4**), iii) MDP_1A group—diabetic rats implanted with the MDP in the subcutaneous space and treated with a single actuation (**0.8 U insulin**) at each of the scheduled times (**n = 4**); and iv) MDP_1A/2A group—diabetic rats implanted with the MDP and treated with a single actuation (**0.8 U insulin**) at each of the scheduled times until 11 days and two consecutive actuations (**1.6 U insulin**) at each of the scheduled times after 11 days (**n = 4**). For each actuation, the external device with M_E was applied and removed to the skin immediately above the implanted MDP. After insulin was delivered via actuation or injection, blood was withdrawn at $T_{\max, \text{insulin}}=60$ min to measure the maximum insulin concentration and also at $T_{\max, \text{glucose}}=120$ min to measure the minimum glucose level and its maximum decrease (Supplementary Fig. 5). **(a,b) The plasma was sampled at scheduled times of 1, 2, 4, 7, 11, 16, 22 and 30 days with the four different animal groups. (c,d) After 30 days, we continued the experiment with the two different animals groups: i) control and iv) MDP_1A/2A groups. At 31 days, for the MDP_1A/2A group, we fully withdrew the insulin solution in the drug reservoir and refilled it with 1.2 ml of a fresh one while the MDP was still implanted (Supplementary Fig. 8). The plasma was sampled at scheduled times of 31, 40, 50 and 60 days.**

Figure 4 Profiles of (a) plasma insulin concentration and (b) blood glucose level at shorter time scales at -1 to 720 min on days 0, 16 and 30, obtained from the four different animal groups: i) control group (**n = 4**), ii) S.C. injection group (**n = 4**), iii) MDP_1A group (**n = 4**) and iv) MDP_1A/2A group (**n = 4**). ** The MDP_1A group was significantly different from both the S.C. injection and MDP_1A/2A groups ($P < 0.05$). * The MDP_1A group was statistically significantly different from the MDP_1A/2A group ($P < 0.05$).

Figure 5 Representative histological images of the tissues around the MDP. Two distinct locations in the tissue were observed: the tissues near the reservoir body surface and near the outlet of the MDP. The asterisk (*) indicates the location of the implanted MDP. **n = 4; 16 days, n = 5; 30 days and n = 4; 60 days. To evaluate biocompatibility, we assessed the degree of inflammatory response in (a) H&E-stained and (b) CD68-stained tissues near both the reservoir body surface and the outlet of the MDP at 30 and 60 days after implantation.** The scale bars are 100 μm . (c) Formation of collagen was assessed in both H&E- and MT-stained tissues near the outlet of the MDP **at 16, 30 and 60 days** after implantation. To measure the capsule thickness, the thinnest region of the capsule was selected in each image of the H&E-stained samples, as indicated by the arrows. The scale bars are 1 mm.

Supplementary Information:

Supplementary Figure 3 Insulin release profiles of the MDP. (a) Figure 2(a) and (b) Figure 2(b) were replotted to show the amount of newly released insulin per actuation. The released

amounts of insulin were highly reproducible and were (a) 0.81 ± 0.04 U and (b) 0.80 ± 0.09 U per actuation. When there was no actuation, insulin was not detected with the measurement method employed in this work.

Supplementary Figure 4 Stability evaluation of insulin. An insulin solution (109 U/ml) in PBS (pH 7.4) was stored **in the MDP** at 37 °C for periods of 0, 10, 20 and 30 days and analyzed by RP-HPLC and circular dichroism (CD) spectroscopy. (a) The RP-HPLC analysis was performed following the same procedure described in Figure 2 in the main text. The uniform intensity and constant retention time of insulin among the tested incubation periods indicate that the insulin **stored in the MDP** was retained without **aggregation or degradation** until day 30 of incubation at body temperature. (b) CD spectra were obtained using a CD spectrometer (J-810, Jasco, Japan). A quartz cuvette with a 1-mm path length was used, and spectra were scanned at 200–250 nm. **The CD spectra were seen to be similar among the tested incubation periods to a large extent, implying that for most of insulin in the MDP, the secondary structure remained unchanged after 30 days of incubation at body temperature.** (c) **A fresh insulin solution and a solution extracted from the MDP after incubation for 30 days were subcutaneously injected to diabetic rats, respectively (n = 4; fresh insulin, n = 4; insulin stored in the MDP for 30 days). A fresh insulin solution was prepared at the same concentration (109 U/ml) as the one initially used for stability test with the MDP and the same volume (7.4 μ l) of each of the solutions was injected. Between two groups, the profiles of blood glucose level were similar, implying that most of insulin stored in the MDP for 30 days could still effectively lower the blood glucose level.**

Supplementary Figure 5 Profiles of plasma insulin concentration and blood glucose level at shorter time scales after insulin administration in diabetic rats. (a) Blood was sampled at -1, 30, 60, 120, 240, 360 and 720 min after insulin administration. The maximum insulin concentration occurred at 60 min in both the S.C. injection and MDP groups. (b) The blood glucose level was measured at -1, 30, 60, 120, 240, 360 and 720 min after insulin administration. The maximum decrease in glucose level occurred at 120 min in both the S.C. injection and MDP groups (**n = 4 ; S.C. injection, n = 4 ; MDP 1A).**

Supplementary Figure 6 Leak test results of the MDP. The plasma insulin concentration was measured immediately before actuation and compared to that of the control group (i.e., diabetic rats without treatment) (**n = 4 ; control, n = 4 ; MDP 1A, n = 4 ; MDP 1A/2A).**

Supplementary Figure 7 Representative images of the intact valve and the one extracted from the MDP biopsied at 60 days after implantation. No sign of clogging was seen with the valve in the implanted MDP.

Supplementary Figure 8 Reproducibility assessment of the MDP after a refilling procedure. For this evaluation, we first simulated consumption of insulin by intentionally withdrawing 0.4 ml of the insulin solution in the drug reservoir and then injected the same volume of fresh insulin solution through the septum port of the MDP using a 30G needle. (a) Schematic description of the refilling procedure. (i) The refill port is located with a localization magnet (M_L) with a donut shape and polarity opposite that of M_R ; (ii) a syringe needle is inserted through the refill port, and **0.4 ml of** insulin solution (109 U/ml) is injected; and (iii) the drug reservoir is filled with the insulin solution. (b) *In vitro* insulin release profiles before and after a refilling procedure. The MDP was fully immersed in phosphate-buffered saline (PBS; pH 7.4) at 37 °C. Initially, three actuations were conducted at 10-min

intervals. After each actuation, the amount of released insulin was measured. Then, a refilling procedure was performed, and the experiments were repeated. **Three MDPs were tested for this experiment.** (c, d) *In vivo* profiles of plasma insulin concentration and glucose level in blood before and after the refilling procedure (**n = 3**). The refilling procedure was performed while the MDP was implanted in STZ-induced diabetic rats. The septum port was found from outside of skin using a localization magnet (M_L) with a polarity opposite that of M_R . To measure the plasma insulin concentration, blood was collected at 60 min after actuation. The blood glucose levels were measured at -1 and 120 min after actuation.

Supplementary Figure 9 Reproducibility assessment of the MDP with varied periods for the external device (M_E) application. Under the *in vitro* drug release experimental condition depicted in Fig. 2, the external device was applied to and removed from the MDP during the periods of < 1 s, 5 s and 10 s, respectively (i.e., MDP (< 1 s), MDP (5 s) and MDP (10 s), respectively). The results revealed that the delivered doses of insulin per actuation were quite similar regardless of the period for the external device application. Three distinct MDPs were tested for each period for the external device application.

Supplementary Figure 10 Profiles of blood glucose level obtained with the three different insulin formulations. In addition to a short acting insulin mainly used in this study (i.e., MDP 1A (short acting insulin)), we additionally prepared the MDPs filled with a rapid acting and long acting insulin formulation (NovoRapid¹ and Lantus², respectively) to give the animal groups of MDP 1A (rapid acting insulin) and MDP 1A (long acting insulin), respectively. At 1 day after the MDP was implanted in STZ-induced diabetic rats, the blood glucose level was obtained at -1 to 720 min after insulin administration by a single actuation of the MDP. A similar dose of insulin was administered for all experimental groups (rapid acting: 0.74 U; short acting: 0.80 U; and long acting 0.74 U). For each animal group, four rats were employed for statistics. For all formulations, the decrease in blood glucose level was apparent after actuation and as expected, a specific profile of blood glucose level was observed for each type of the insulin formulations. For rapid acting insulin, the blood glucose level decreased and increased back more rapidly. For long acting insulin, the blood glucose level dropped relatively slowly and this lowered level was maintained for a longer period.

Supplementary Figure 11 *In vivo* profiles of blood glucose level with multiple daily actuations of the MDP. To give an insight of insulin delivery after each of the meal times, the MDP was actuated twice with an interval of 12 h. After the first actuation, a glucose level dropped and increased back to a high level as expected with diabetic rats, which was observed to be repeated in a similar pattern after the second actuation (n=4).

Reviewers' comments:

Reviewer #1 (Remarks to the Author):

The authors have addressed the questions that I raised at 1st review satisfactorily.

Reviewer #2 (Remarks to the Author):

The authors have responded to previous review comments. Since this work will interests mostly technologists, I would also suggest that the authors cite the following relevant recent work. Current literature review focus mostly on insulin pump related work. But as MEMS drug delivery research is growing as a research field, it would make sense to acknowledge previous work in the area and distinguish this work from state-of-the-art MEMS drug delivery devices.

On-demand drug delivery from local depots Y Brudno, DJ Mooney - Journal of Controlled Release, 2015 <http://dx.doi.org/10.1016/j.jconrel.2015.09.011>

Shi, Jingru and Zhang, Hongbin and Jackson, John and Shademani, Ali and Chiao, Mu, A robust and refillable magnetic sponge capsule for remotely triggered drug release Journal of Materials Chemistry B v4, n46, pages, 7415–7422, 2016

Refillable and magnetically actuated drug delivery system using pear-shaped viscoelastic membrane

H So, YH Seo, AP Pisano - Biomicrofluidics, 2014 Aug 25;8(4):044119.

MEMS-enabled implantable drug infusion pumps for laboratory animal research, preclinical, and clinical applications E Meng, T Hoang - Advanced drug delivery reviews,

Reviewers' comments:

Reviewer #1 (Remarks to the Author):

The authors have addressed the questions that I raised at 1st review satisfactorily.

We do appreciate for the affirmative comments.

Reviewer #2 (Remarks to the Author):

The authors have responded to previous review comments. Since this work will interests mostly technologists, I would also suggest that the authors cite the following relevant recent work. Current literature review focus mostly on insulin pump related work. But as MEMS drug delivery research is growing as a research field, it would make sense to acknowledge previous work in the area and distinguish this work from state-of-the-art MEMS drug delivery devices.

On-demand drug delivery from local depots Y Brudno, DJ Mooney - Journal of Controlled Release, 2015 <http://dx.doi.org/10.1016/j.jconrel.2015.09.011>

Shi, Jingru and Zhang, Hongbin and Jackson, John and Shademani, Ali and Chiao, Mu, A robust and refillable magnetic sponge capsule for remotely triggered drug release Journal of Materials Chemistry B v4, n46, pages, 7415–7422, 2016

Refillable and magnetically actuated drug delivery system using pear-shaped viscoelastic membrane

H So, YH Seo, AP Pisano - Biomicrofluidics, 2014 Aug 25;8(4):044119.

MEMS-enabled implantable drug infusion pumps for laboratory animal research, preclinical, and clinical applications E Meng, T Hoang - Advanced drug delivery reviews,

We appreciate for the useful comments from the reviewer. We have added the references and revised the text in the manuscript as suggested.

Page 3

Therefore, a more patient-friendly system with controlled delivery of insulin has been actively sought. After one-time implantation (or injection), such a system must permit pulsatile insulin release to mimic physiological secretion following food intake⁵. **More importantly, this release pattern must be controlled in a patient-driven, on-demand manner from outside the body without invasive multiple skin punctures**⁶. Stimulus-

responsive biomaterials previously used to deliver insulin⁷⁻⁹ are subject to ruptures, leaks or deformations that lead to a lack of reproducibility over multiple release cycles. More sophisticated, implantable microdevices operated by the principles of peristaltic actuation^{10,11} electrochemical dissolution^{12,13}, electrothermal ablation^{14,15}, electrolysis^{16,17} or piezoelectric actuation¹⁸ **often require electrical power supplies (e.g., batteries) and electronic circuit components**¹⁹ and are thus too large for implantation. Moreover, when the battery expires, device explantation is inevitable and requires additional major surgery, making this method unsuitable for long-time insulin delivery.

Page 10

In other examples, a membrane with an aperture or porous capsule was deformed to squeeze the drug depot and initiate its release by a magnetic field³⁰⁻³².

References

6. Brudno, Y. & Mooney, D.J. On-demand drug delivery from local depots. *J. Control. Release* 219, 8-17 (2015).
19. Meng, E. & Hoang, T. MEMS-enabled implantable drug infusion pumps for laboratory animal research, preclinical, and clinical applications. *Adv. Drug Deliv. Rev.* 64, 1628-1638 (2012).
31. So, H., Seo, Y.H. & Pisano, A.P. Refillable and magnetically actuated drug delivery system using pear-shaped viscoelastic membrane. *Biomicrofluidics* 8, 044119 (2014).
32. Shi, J., Zhang, H., Jackson, J., Shademani, A. & Chiao, M. A robust and refillable magnetic sponge capsule for remotely triggered drug release. *J. Mater. Chem. B* 4, 7415-7422 (2016).

Reviewers' Comments:

Reviewer #2 (Remarks to the Author)

The authors have addressed my previous comments. I recommend publication of this article.